# Pre-mitotic genome re-organisation bookends the B cell differentiation process

Wing Fuk Chan[1,2,4], Hannah D. Coughlan[1,2,4], Jie H. S. Zhou[1,2], Christine R. Keenan [1,2], Naiara G. Bediaga[1,2], Philip D. Hodgkin[1,2], Gordon K. Smyth [1,3,4], Timothy M. Johanson[1,2,4] & Rhys S. Allan [1,2,4✉]

During cellular differentiation chromosome conformation is intricately remodelled to support the lineage-specific transcriptional programs required for initiating and maintaining lineage identity. When these changes occur in relation to cell cycle, division and time in response to cellular activation and differentiation signals has yet to be explored, although it has been proposed to occur during DNA synthesis or after mitosis. Here, we elucidate the chromosome conformational changes in B lymphocytes as they differentiate and expand from a naive, quiescent state into antibody secreting plasma cells. We find gene-regulatory chromosome reorganization in late G1 phase before the first division, and that this configuration is remarkably stable as the cells massively and rapidly clonally expand. A second wave of conformational change occurs as cells terminally differentiate into plasma cells, coincident with increased time in G1 phase. These results provide further explanation for how lymphocyte fate is imprinted prior to the first division. They also suggest that chromosome reconfiguration occurs prior to DNA replication and mitosis, and is linked to a gene expression program that controls the differentiation process required for the generation of immunity.

[1] The Walter and Eliza Hall Institute of Medical Research, Parkville, VIC, Australia. [2] Department of Medical Biology, The University of Melbourne, Parkville, VIC, Australia. [3] School of Mathematics and Statistics, The University of Melbourne, Parkville, VIC, Australia. [4]These authors contributed equally: Wing Fuk Chan, Hannah D. Coughlan, Gordon K. Smyth, Timothy M. Johanson, Rhys S. Allan. ✉email: rallan@wehi.edu.au

Extraordinary three-dimensional (3D) genome organization is critical to simply fit complex genomes into the nuclear space. However, this organization also plays a critical role in supporting gene expression. As such, DNA loops formed between gene promoters and distant regulatory elements such as enhancers are critical for the appropriate expression of most genes, while complex genomes are divided into transcriptional units known as topologically associated domains (TADs)[1–3]. Recent work has shown that these chromosome structures can be altered to drive cell type-specific gene-regulatory programs[4–6]. It is unknown how such ornate structure is established and maintained through the rigors of cell proliferation and differentiation while continuing to support cell identity. As such, progression through the cell cycle exposes chromosomes to a number of biophysical challenges, such as DNA replication and mitosis, that potentially impact the architecture required for gene regulation. Recent studies using chromosome conformation capture techniques demonstrated dynamic and cell cycle stage-specific genome organization in cells undergoing normal turnover[7,8]. These studies found that chromosomes become highly condensed in preparation for mitosis, rapidly expand in early G1, before remaining relatively stable until undergoing extensive remodeling during replication[7,8]. However, to date, the dynamics of genome-wide chromosome architecture across cell cycle and division when cells are exposed to differentiation signals are unclear. It has been proposed that DNA replication is a time when chromosome architecture could be remodeled[9–11]. While other groups have suggested that the hours after mitosis represent an opportunity for chromosome reorganization[12–17].

Naive B cells reside in the secondary lymphoid organs in a quiescent state (G0 of the cell cycle). During the initiation of an immune reaction, they process activation signals from pathogens and immune accessory cells before entering into the proliferative response phase. Compared to time spent in subsequent divisions, the cellular events leading up to the first post-activation division take a relatively long time. Within this first cell cycle after activation, cells spend a majority of time in the G1 phase, prior to shorter times in the S phase, G2 phase, and then mitosis, which occurs ~30 h after the initial stimulus[18,19]. Evidence suggests that fate decisions of T and B lymphocytes are imprinted in this prolonged first post-activation G1 phase and then faithfully transmitted to clonal descendants[19–24]. Subsequently, activated B cells undergo rapid cell proliferation, dividing approximately every 8–10 h, with very short G1 phases. This process of massive and rapid cell proliferation is essential for the clonal expansion that underlies the antigen-specific immune response[25]. These activated B cells differentiate in a division-linked fashion into antibody-secreting plasma cells and also form long-lived memory cells, which are able to rapidly become antibody-secreting cells upon rechallenge[26,27]. A number of recent studies, including our own, have compared the chromatin structure of resting and activated or differentiated B cells and identified large-scale activation-induced changes in 3D genome organization[5,6,28–33]. However, these studies did not investigate when in the developmental process the genome was remodeled.

Here, by performing a fine spatiotemporal analysis of B-cell activation, we reveal that the changes to 3D chromatin structure occur in two discrete windows, associated with prolonged time in the G1 phase of the cell cycle. Overall, we propose that the 3D genome is reconfigured in response to differentiation signals prior to DNA synthesis and mitosis to ensure the implementation of a transcriptional program required for the generation of B-cell immunity.

## Results

**Waves of genome reorganization bookend plasmablast differentiation.** Previously, we have found that naive B cells substantially alter their genome organization as they become differentiated plasmablasts[34]. To further understand when these changes arose and their association with transcriptional programs here, we examine naive B cells, B cells immediately following activation with lipopolysaccharide (3 and 10 h post-activation), those just prior to the first division (33 h, imminent division), the massively expanded B-cell population (96 h post-activation) and, finally, terminally differentiated plasmablasts (Fig. 1a and Supplementary Fig. 1A–E). We then used differential RNA-Seq and in situ HiC analyses between activation conditions with two biological replicates per condition (Supplementary Data 1) to reveal gene expression and 3D genome changes, respectively. Transcriptomic analysis, using the quasi-likelihood (QL) framework[35,36] of edgeR, revealed that the majority of differentially expressed genes (DEs) occur prior to the first division (Fig. 1b, c and Supplementary Fig. 1F, Supplementary Data 2). As such, the expression of 5838 genes is significantly altered in the first 3 h post-activation, a further 3963 in the following 7 h, and a further 1877 before the first division. A comparatively small number of transcriptional changes (1371 DEs) are observed as B cells clonally expand followed by a second wave of transcriptional change (3340 DEs) marking differentiation into plasmablasts. After determining the HiC libraries were reproducible between biological replicates of the same condition and of sufficient quality (Supplementary Data 1, Supplementary Fig. 1G, H), significant changes in genome organization, or differential interactions (DIs) were determined between samples with the diffHic pipeline[37]. diffHic partitions the genome into bins, counts the number of read pairs mapping to each pair of bins, then tests each bin pair for significant changes in intensity between stages, using the methods in the edgeR package[38] which incorporate the biological variability into the modeling (Supplementary Fig. 1G, H). In contrast to the pattern of transcriptional change, in situ HiC and diffHic reveal very few genome organizational changes in the first 10 h after activation (Fig. 1d, e). In fact, the major genome organizational changes occur in two distinct waves; the first between 10 and 33 h, prior to the first cell division (10,628 differential interactions; DIs), and the second upon plasmablast differentiation (6784 DIs) (Fig. 1d, e and Supplementary Data 3).

The 3D structure around the Twistnb gene, which encodes a component of the RNA polymerase I complex, models the first wave of change with structure diminishing in the first wave of organizational change, presumably when the RNA polymerase I complex is no longer required. As such, increased 3D contacts between the Twistnb promoter and distal sites in the genome (potentially enhancer elements) are detected prior to the first activation-induced division but not after (Fig. 1f). In contrast, DNA structure around the Bcl6 gene models the second wave of organizational change, with relatively stable long-range interactions occurring both pre- and post-activation until plasmablast differentiation when this structure is lost (Fig. 1f). The organizational changes around the Bcl6 gene, among others, such as Ebf1, Prdm1, and Id2 (Supplementary Fig. 1I), are linked to their expression pattern, suggesting that chromosome structure potentially has a role in regulating Bcl6 expression, as has been previously suggested in human B cells[29].

In addition to identifying highly stage-restricted waves of organizational change, one of which occurs prior to the first cell division, our examination of the 3D structure during B-cell differentiation highlights two points. The first is that given the relative absence of early activation-induced genome organizational changes, the rapid and dramatic transcriptional changes that occur immediately post-activation (Fig. 1b, c) are either driven independently of 3D structure or rely on pre-existing structures. Indeed, analysis of chromatin loops in naive B cells revealed a significant correlation between pre-existing loop structure and gene expression changes 3 h after activation

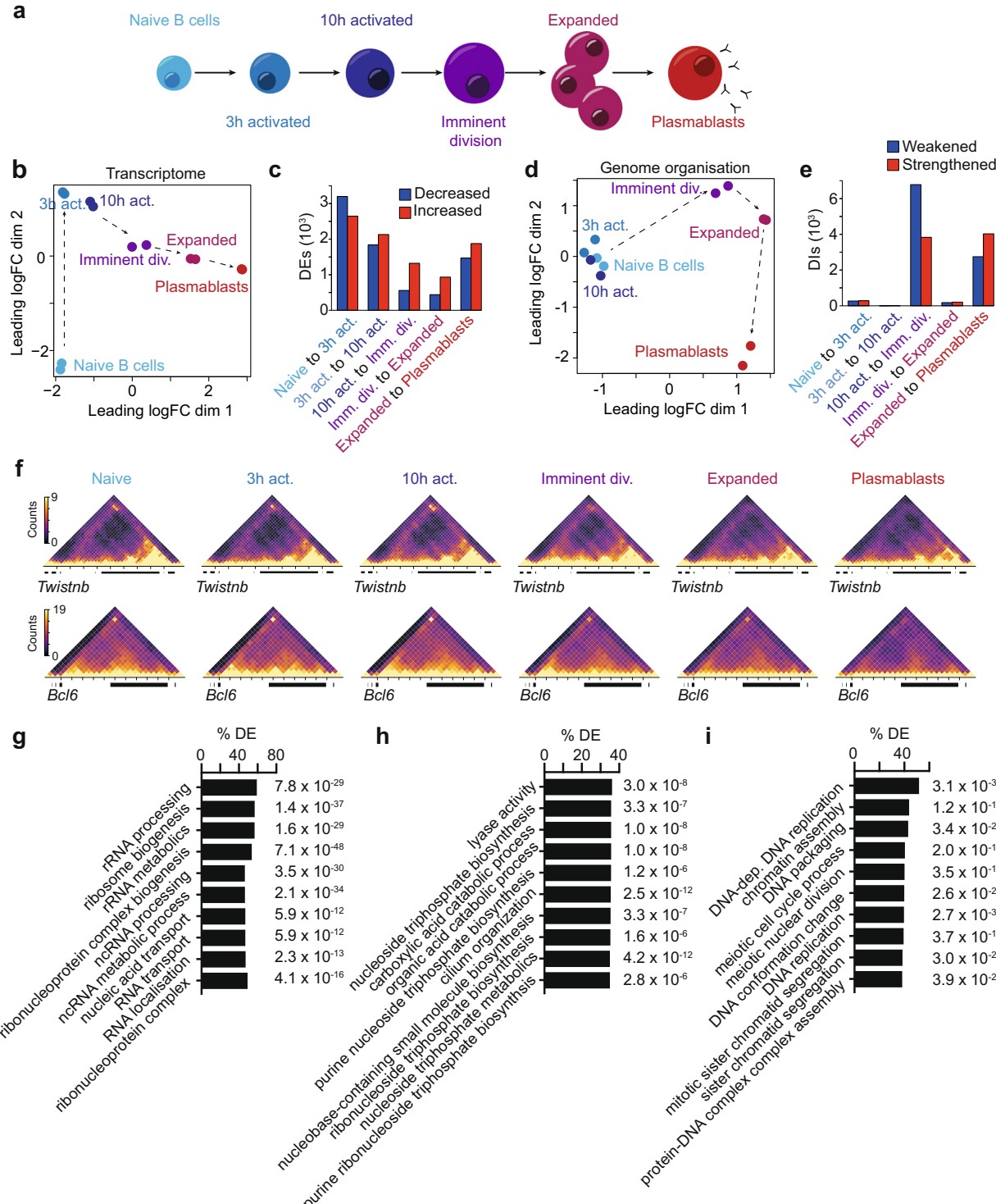

**Fig. 1 Waves of genome reorganization bookend plasmablast differentiation. a** Stages of B-cell activation and differentiation. **b** Multidimensional-scaling plot of RNA-seq data of B-cell activation. **c** Numbers of differentially expressed genes between B-cell activation stages calculated with fold changes significantly above 1.5 (Treat FDR < 0.05). Blue indicates a decrease in expression level with activation. Red indicates an increase in expression level with activation. **d** Multidimensional-scaling plot of B-cell activation demonstrating two dominant waves of genome organizational change. **e** Numbers of differential interactions (DIs) between B-cell activation stages. Blue indicates the weakening of organization with activation. Red indicates strengthening of organization with activation. **f** Normalized in situ HiC contact matrices at 50 kbp showing genome organization at the *Twistnb* (chr12:33.2–35.3 Mb) and *Bcl6* (chr16:23.85–25.15 Mb) loci at each stage of B-cell activation. Color scale indicates the number of reads per bin pair. **g** Gene ontological terms with genes increasing in differential expression between the naive and 3 h activated B cells over-represented. Groups ranked according to the percentage of groups genes that are differentially expressed. **h** Gene ontological terms with genes increasing in differential expression between the 3 and 10 h activated B cells over-represented. Groups ranked according to the percentage of genes that are differentially expressed. **i** Gene ontological terms with genes increasing in differential expression between the 10 h activated B cells and those imminently to divide over-represented. Groups ranked according to the percentage of groups genes that are differentially expressed. Source data are provided as a Source Data file.

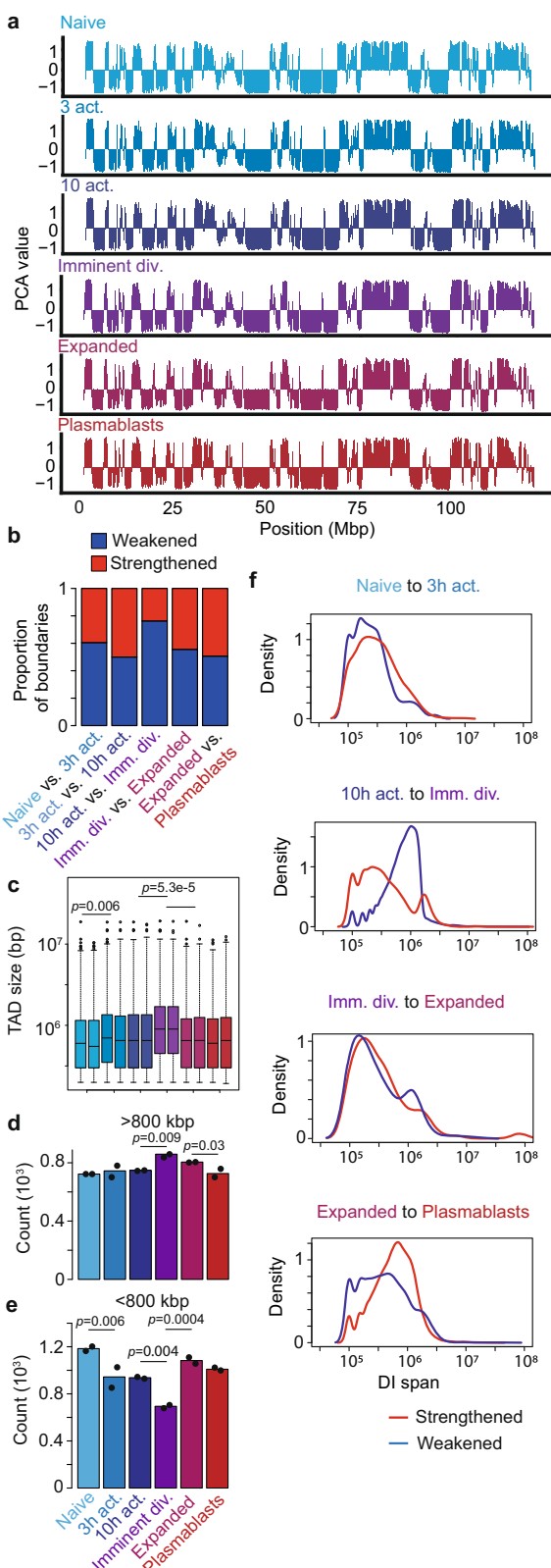

**Fig. 2 Large genomic structures are altered by early B-cell activation and mitosis. a** A/B compartmental interaction plots of in situ HiC data of the entirety of chromosome 12 at different B-cell activation stages. 100 kB resolution. **b** Proportion of TAD boundaries strengthening (red) or weakening (blue) at each transition of B-cell activation determined by diffHic. **c** 1–99th percentile boxplot of distributions of TAD size (bp) from replicate libraries called with TADbit at each stage of B-cell activation. A one-way ANOVA was used to determine significance. **d**, **e** Number of TADs detected that are larger (**d**) or smaller (**e**) than 800 kbp at each stage of B-cell activation in two biologically independent libraries. One-way ANOVAs were used to determine significance. **f** Distribution of DI spans (distance between anchors of significance DIs) for strengthened and weakened interactions at each B-cell differentiation transitions during B-cell differentiation (50 kbp resolution). Blue indicates DIs that are weakened (intensity fold-change < 1). Red indicates DIs that are strengthened (fold-change > 1). Source data are provided as a Source Data file.

(Fig. 1g and Supplementary Data 5), likely reflecting the B cells immediate response to activation. Between 3 and 10 h, we observed enrichment for metabolic genes (Fig. 1h and Supplementary Data 5), while, of note, the predicted function of the transcriptional changes in the hours just prior to the first cell division include chromatin remodeling and DNA conformation change (Fig. 1i and Supplementary Data 5). The second point of interest is the relatively small number of organizational changes that occur as B cells clonally expand (Fig. 1d, e). This suggests that the genomic structures that B cells create prior to the first division are preserved over numerous subsequent divisions and could account for the striking familial symmetry in fates observed by long-term imaging[18,39]. This may reflect the time restrictions imposed by rapid clonal expansion.

**Large genomic structures are altered by early B-cell activation and mitosis**. We next examined the impact of B-cell activation on large-scale genome organization. Assigning the genome at each transition into A/B compartments, we find little to no change in large-scale compartmentalization (Fig. 2a and Supplementary Data 6). In contrast, we find significant changes to TAD structure during B-cell differentiation using two independent analysis methods. One uses TADbit[40] to segment chromosomes into TADs in individual samples while the second uses diffHic to identify putative TAD boundaries that strengthen and weaken between B-cell differentiation stages. diffHic finds TAD boundaries are generally weakened just prior to the first division (Fig. 2b) while TADbit finds that TADs become larger (Fig. 2c) and less numerous (Fig. 2d, e) as a consequence at the same time point. This is likely reflective of the loss of TAD structure known to be associated with chromosome condensation prior to mitosis[7,8]. An examination of individual DIs confirms that it is the longer-range interactions that tend to be weakened prior to the first division (Fig. 2f). A similar, albeit weaker, pattern of diminishing TADs is also observed in the first 3 h after B-cell activation (Fig. 2b–f), possibly reflecting the transcriptional and cellular responses to activation.

**Gene promoter interactivity correlates with transcription**. To examine the relationship between organizational changes and alterations in gene expression, we developed a promoter-focused strategy to interrogate our organizational data. Put simply, we set a 10 kb window around all protein-coding promoters and identify all DNA–DNA interactions that exist between the promoter and any other genomic region (Fig. 3a). Using the statistical framework of the R package edgeR, we determine significant differences

($p = 2.46 \times 10^{-6}$) (Supplementary Data 4). The lack of early changes in genome organization may be explained by the transcriptional focus of the cell at these times. For example, in the first 3 h post-activation, gene ontological analysis of transcriptional changes (Supplementary Fig. 1J, K) revealed that genes involved in ribosome biogenesis are significantly enriched

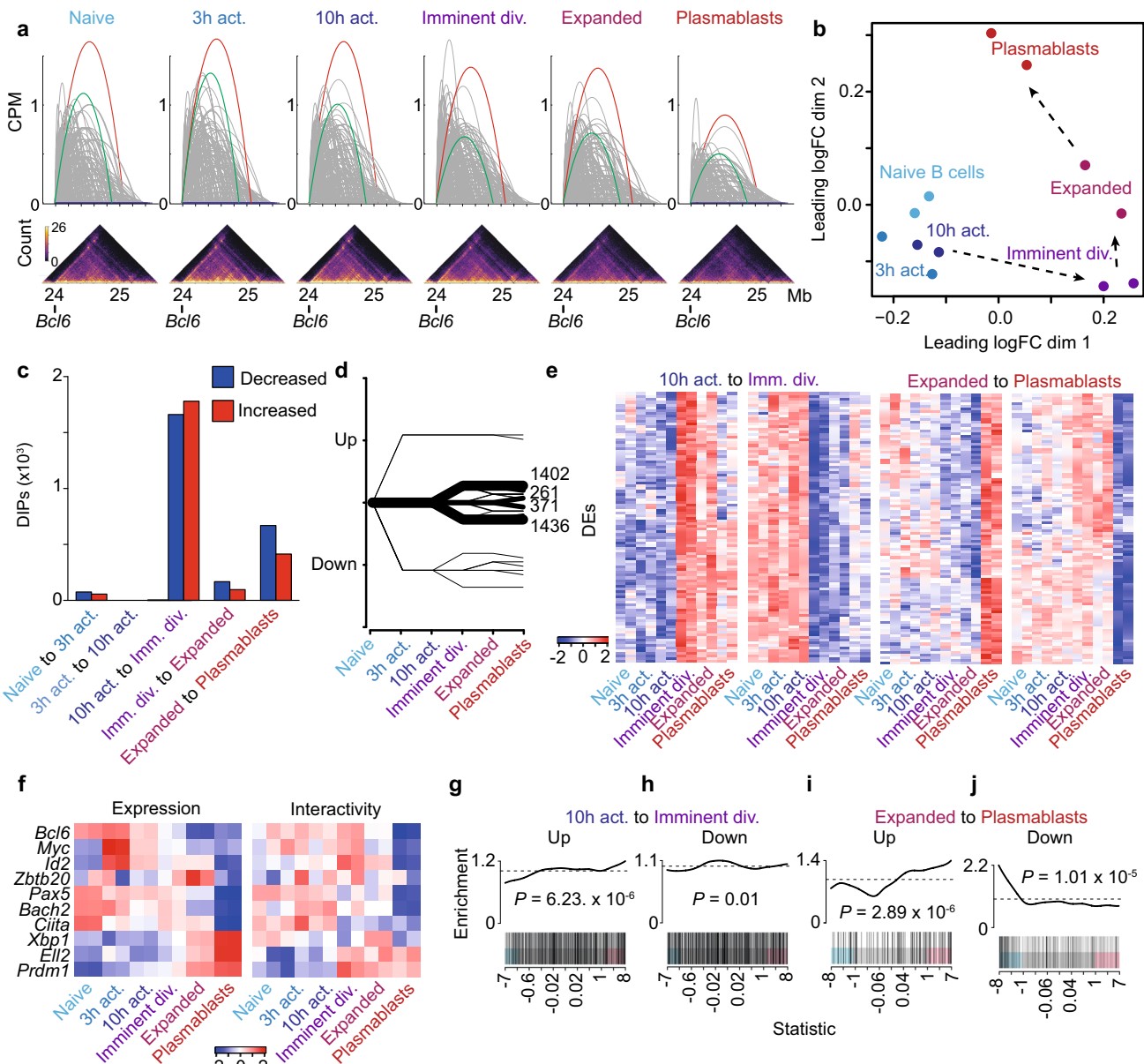

**Fig. 3 Gene promoter interactivity reveals a direct link between reorganization and transcription. a** Schematic showing how a promoter interaction count is determined. Each interaction with the *Bcl6* promoter is shown as an arc. The sum of these interactions in counts per million (CPM) is the promoter interaction count. The red and green arcs show select interactions that represent dominant looping interactions. Below are the corresponding raw in situ contact matrices (10 kbp resolution). Color scale indicates the number of reads per interaction. **b** Multidimensional-scaling plot of the filtered and normalized logCPM values for interacting promoters during B-cell activation. **c** Bar plot showing the number of differentially interacting promoters (DIPs) between each transition of B-cell activation. Blue indicates decreased interactivity. Red indicates increased interactivity. **d** Plot showing the patterns of change and numbers of DIPs during B-cell activation. Line thickness represents the number of DIPs in each pattern. **e** Heatmap of logCPM of the top 100 DIPs by false discovery rate in patterns of change in DIPs. The patterns shown represent the two dominant waves of organizational change, either exclusively increasing or decreasing at the 10 h activation to imminent division transition or the expanded to plasmablasts transition. **f** Heatmaps showing a change in expression (logRPKM) and change in promoter interactions (logCPM) in key B-cell differentiation genes across B-cell activation. **g** Barcode enrichment plot showing the correlation between promoters that significantly increase in promoter interactions as B cells prepare to divide (10 h activation to imminent division) with increases in gene expression at the same transition (Fig. 1). Genes are ordered on the plot from right to left (x axis) from most upregulated to most downregulated according to the moderated *F*-statistic. *p*-value was calculated with the fry test. **h** Barcode enrichment plot showing the correlation between promoters that significantly decrease in promoter interactions as B cells prepare to divide (10 h activation to imminent division) with decreases in gene expression at the same transition (Fig. 1). Genes are ordered on the plot from right to left (x axis) from most upregulated to most downregulated according to the moderated *F*-statistic. *p*-value was calculated with the fry test. **i** Barcode enrichment plot showing the correlation between promoters that significantly increase in promoter interactions as B cells differentiate into plasmablasts (expanded to plasmablast) with increases in gene expression at the same transition (Fig. 1). Genes are ordered on the plot from right to left (x axis) from most upregulated to most downregulated according to the moderated *F*-statistic. *p*-value was calculated with the fry test. **j** Barcode enrichment plot showing the correlation between promoters that significantly decrease in promoter interactions as B cells differentiate into plasmablasts (expanded to plasmablast) with decreases in gene expression at the same transition (Fig. 1). Genes are ordered on the plot from right to left (x axis) from most upregulated to most downregulated according to the moderated *F*-statistic. *p*-value was calculated with the fry test. Source data are provided as a Source Data file.

in promoter interactivity between differentiation stages (Supplementary Fig. 2A–D). These significantly altered promoters are known as Differentially Interacting Promoters (DIPs). Simply plotting DIPs between all six stages of B-cell differentiation, we find that DIPs separate the populations similarly to DEs or DIs (Fig. 3b and Supplementary Data 7). Furthermore, examining the distribution of DIPs across all transitions, we find, similar to DIs, they exist in two distinct waves; imminently before the first division and at the final transition to plasmablast (Fig. 3c and Supplementary Fig. 2E, F, Supplementary Data 7). In addition to examining changes at each differentiation transition, we group DIPs based on promoter interaction changes across the six stages of differentiation. This pattern analysis reveals 20 distinct patterns (Supplementary Data 8). However, the majority of DIPs can be grouped into just four patterns; increased or decreased at the imminent division population then remaining unchanged at all further transitions (2838 DIPS), or increased or decreased only upon terminally differentiating into plasmablasts (632 DIPs) (Fig. 3d, e and Supplementary Fig. 2G). Examining our HiC data in this way highlights the stability but also the specificity of gene promoter interactions during B-cell differentiation. Examining the interactivity of key B-cell genes, we find interactivity reflects expression (Fig. 3f). For example, similar to their patterns of expression, the promoters of Prdm1 and Ell2 show increasing levels of interaction as B-cell differentiation progresses. Conversely, the promoters of Bcl6, Myc, Id2, Pax5, and Bach2 show marked loss of interactions at the plasmablast stage, showing striking similarity to their patterns of expression (Fig. 3f). A broader analysis of gene ontology of DIP genes, both at each transition and within dominant patterns of change, finds that the related biological processes fit with those associated with the particular stage of differentiation (Supplementary Data 9 and 10). For example, promoters that showed significant and stable increases in the interaction between 10 h post-activation though prior to the first division associate with cell cycle and B-cell proliferation, among others.

While select genes appear to show a positive association between expression and organization, expanding our analysis to include all genes reveals a significant correlation between the DIPs at the two major waves of promoter interaction change and expression of the associated genes (Fig. 3g–j). For example, genes that increase expression between 10 h post-activation but prior to the first division are significantly correlated with DIPs that also increase. The same pattern is observed in DEs during plasmablast differentiation (Fig. 3i, j).

**Identifying transcription factor motifs that are associated with the reorganization of the genome during B-cell differentiation**. A number of groups have used gene-regulatory networks to understand the DNA-binding transcription factors that govern B-cell development and differentiation[27,40–42]. More recent work has also defined the gene-regulatory network in the context of the 3D genome in activated B cells[30]. To continue to catalog the DNA-binding proteins within the 3D genome throughout B-cell differentiation, we determined the prevalence on DNA-binding motifs of expressed transcription factors (>1 RPKM) within the distant regions that interact with DIPs, relative to long-range interacting elements that are not differential during B-cell differentiation.

This analysis revealed the motifs of 160 transcription factors enriched within DIPs across B-cell differentiation ($p$-value < 0.05, adjusted for multiple tests using a Bonferroni correction) (Supplementary Data 11), hinting that many more transcription factors than previously demonstrated may regulate immune cell gene expression and these changes are linked to alterations in 3D

genome organization[34,43]. These include factors known to regulate B-cell differentiation (Pax5, Bach2, Irf4, Irf8, and Blimp1) (Fig. 4a). Seventeen of these transcription factor motifs are enriched in DIPs at only one transition of B-cell differentiation (Fig. 4b–d), suggesting they may perform a stage-specific function. For example, consistent with its known function as a regulator of B-cell proliferation[44], the motif of Hif1a is specifically enriched in DIPs with decreasing interactivity just prior to the first division. As B cell clonally expand and begin the process of differentiation, we find that the motif of Prdm1 (Blimp1), consistent with its expression pattern (Supplementary Data 4) and known role as a repressor of B-cell genes[45,46] is enriched specifically in DIPs with decreasing interactivity (Fig. 4c).

To further explore the potential function of the DIP-associated transcription factors, we examined their expression level alongside their motif association with increased or decreased interactivity (Fig. 4e, f). We hypothesize that transcription factors with increased expression at a transition where their motif is enriched in DIPs with increasing interactivity are likely transcriptional activators. Conversely, factors that decrease in expression at a transition but with enriched motifs in increasing DIPs are possibly functionally repressive. This analysis identifies numerous putative activators and repressors that function during the two major waves of organizational change (Fig. 4e, f). For example, we find that Foxm1 increases in expression and motif association with increased DIPs just prior to the first division. This is consistent with its known role as a positive regulator of cell division in B cells[47]. Conversely, the transcriptional repressor Gfi1[48] is upregulated and specifically motif enriched in DIPs with decreasing interactivity as B cells differentiate (Fig. 4f).

E2f7 and Pax5 are two other TFs that show large changes in an expression just prior to cell division and plasmablast differentiation, respectively. These transcription factors are known to perform diverse stage-specific functions during B-cell differentiation, with E2f7 a well-known repressor of cell cycle early in B-cell differentiation[49] and Pax5 known to activate B-cell-specific genes while repressing plasma cell genes[40]. To explore potential diversity or consistency of stage-specific function in the motif-enriched TFs during B-cell differentiation, we compared the transcription factors expression changes as B cells prepare to first divide and then to differentiate (Fig. 4g, h). Again, assuming that increased expression associated with the presence of a TF motif in DIPs indicates a potential role as an activator, we find a pattern that suggests transcription factors can be consistently associated with activation, repression, or both. As expected, Pax5 and E2f7 display these dual associations (Fig. 4g, h). However, consistent with their known functions, factors such as Rfx1 and Irf4 are consistently associated with repression[50,51] while factors such as Klf4 are consistently associated with activation[52,53] (Fig. 4g, h). To further explore the association of Irf4 with repression, we overlaid chromatin-immunoprecipitation (ChIP) data for Irf4 in activated B cells and plasma cells[46] with DIPs at these transitions. We find that Irf4 binding sites are significantly enriched within the regions that interact with the promoters of genes with decreased interactivity compared to those with increased interactivity, at both transitions ($p = 1.75 \times 10^{-31}$ and $p = 3.67 \times 10^{-30}$, respectively). Similarly, ChIP data for Blimp1 binding in plasmablasts[45] finds Blimp1 significantly enriched ($p = 3.83 \times 10^{-35}$) in the regions of the genome that link with the promoters of genes with decreased interactivity, compared to those with increased interactivity. Thus, consistent with our association analyses and those of others, the presence of Irf4 or Blimp1 within promoter interacting regions significantly increases the chance that these structures are altered.

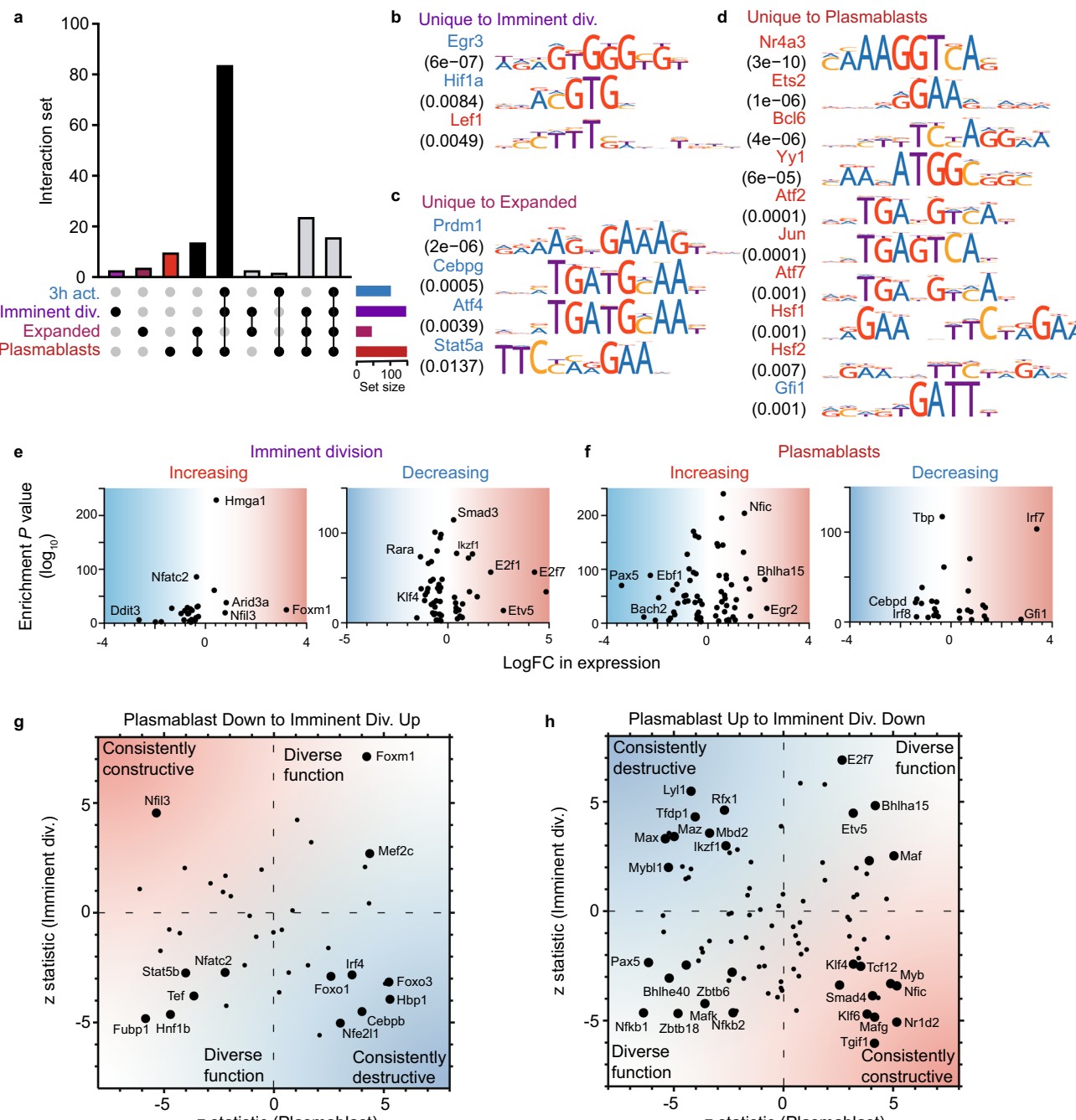

**Fig. 4 Numerous transcription factors regulate B-cell differentiation via 3D genome organization. a** Upset plot showing shared and unique enriched motifs in DIP patterns that are exclusively increasing or decreasing at the transitions. The vertical bars indicate the number of motifs shared between the patterns and the black dots indicate the patterns shared. The horizontal bars denote the number of motifs enriched in the pattern. **b** Motifs enriched uniquely in the exclusively increasing or decreasing 10 h activation to imminent division DIP patterns ranked by adjusted p-value. The color of the motif indicates whether it was enriched in the increasing (red) or decreasing (blue) DIPs pattern. **c** Motifs enriched uniquely in the exclusively increasing or decreasing imminent division to expanded patterns ranked by adjusted p-value. **d** Motifs enriched uniquely in the exclusively increasing expanded to plasmablasts patterns ranked by adjusted p-value. **e** The adjusted p-value (−log10) of the motifs enriched in the exclusively increasing or decreasing 10 h activation to imminent division DIP patterns plotted as a function of logFC of expression at that transition. Blue shading represents decreased expression, red represents increased expression. **f** The adjusted p-value (−log10) of the motifs enriched in the exclusively increasing or decreasing expanded to plasmablasts DIP patterns plotted as a function of logFC of expression at that transition. Blue shading represents decreased expression, red represents increased expression. **g** The z-statistic of expression for the 10 h activation to imminent division transition plotted against the z-statistic of expression for the expanded to plasmablasts. The genes plotted correspond to motifs enriched in the exclusively increasing 10 h activation to imminent division DIP pattern and exclusively decreasing expanded to plasmablasts DIP pattern. Labeled genes are significant in both transitions (FDR < 0.05). Blue shading represents potentially destructive factors, red represents potentially constructive factors. **h** The z-statistic of expression for the 10 h activation to imminent division transition plotted against the z-statistic of expression for the expanded to plasmablasts. The genes plotted correspond to motifs enriched in the exclusively decreasing 10 h activation to imminent division DIP pattern and exclusively increasing expanded to plasmablasts DIP pattern. Labeled genes are significant in both transitions (FDR < 0.05). Blue shading represents potentially destructive factors, red represents potentially constructive factors. Source data are provided as a Source Data file.

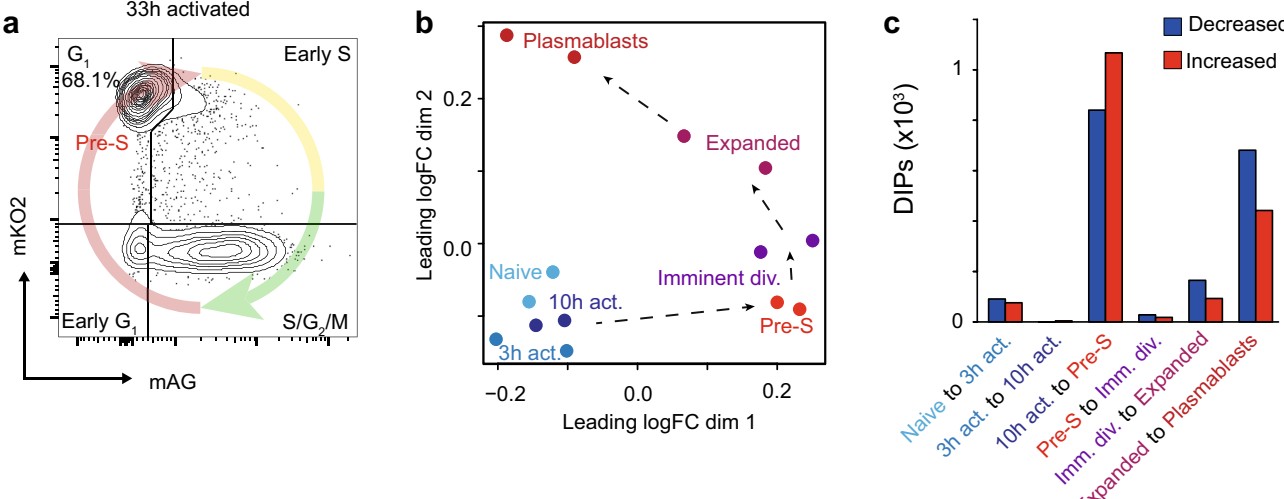

**Fig. 5 Genome organizational changes occur prior to DNA synthesis. a** Flow cytometry plot of $G_1$ stage FUCCI B cells that were flow cytometry purified by monomeric Kusabira Orange 2 (mKO2) and monomeric Azami Green (mAG) expression just prior to the first division (33 h). **b** Multidimensional-scaling plot of the filtered and normalized logCPM values for interacting promoters during B-cell activation with Pre-S samples included. **c** Bar plot showing the number of differentially interacting promoters (DIPs) between each transition of B-cell activation with Pre-S samples included. Blue indicates decreased promoter interactions. Red indicates increased promoter interactions. Source data are provided as a Source Data file.

Taken together these analyses suggest that overlays of transcription factor expression data and genome organizational data can be used to reveal further TFs that may influence B-cell activation and differentiation.

**Genome organizational changes before the first division occur prior to DNA synthesis.** Our data identifies that the genome architecture that defines the clonally expanded population is established prior to the first mitosis. However, it not clear if this genome reorganization occurs throughout interphase or at a specific stage of the cell cycle. To address this issue, we utilized the Fluorescent Ubiquitination-based Cell Cycle Indicator (FUCCI) system[54] to isolate imminently dividing B cells specifically in the G1 stage of the cell cycle (Fig. 5a and Supplementary Fig. 3A). While imminently dividing, these cells are yet to enter the DNA synthesis phase. Using a DIPs analysis to compare this pre-S phase population to the bulk imminently dividing B cells and other stages of differentiation allows greater dissection of the contribution of cell-cycle dependent changes on the observed DIs. In doing so, we find that the majority of the organizational changes that occur prior to the first division occur prior to S-phase (Fig. 5b, c and Supplementary Data 12), demonstrating that genome reorganization occurs prior to DNA synthesis.

**Major genome organizational change transitions associate with prolonged time in the G1 phase.** Given that the first wave of genome reorganization occurs in a prolonged G1 phase of the cell cycle, we next examine whether the second wave of organizational change is associated with a prolonged G1 phase. As such, using the FUCCI system to examine the final stage of B-cell differentiation, we find that as B cells prepare to differentiate into plasmablasts (day 3 of differentiation versus day 5) a greater proportion are found in G1, consistent with spending increasing amounts of time at this stage of the cell cycle (Fig. 6a). In contrast, once differentiated, plasmablasts appear to spend significantly less time in G1 (Fig. 6b and Supplementary Fig. 3B). Thus, again we observe a naturally prolonged G1 phase associated with major genome organizational change. Given this association, we next test whether artificially lengthening particular phases of the cell cycle could promote differentiation (Fig. 6c). We show that

treatment of CD40L/IL-4 stimulated Blimp1-GFP naive B-cell cultures with Purvalonol A (a cyclin-dependent kinase inhibitor that prolongs the G1 phase of the cell cycle[55]) drives significantly more plasmablast differentiation than treatment with L-mimosine [an L-alpha-amino acid that disrupts DNA synthesis and prolongs the S phase of the cell cycle[56]] or no treatment (Fig. 6d, e). These results are in line with our previous data[57] and supports the idea that the time spent in the G1 phase of the cell cycle might be limiting and particularly important to allow cell fate decisions and genome organizational changes to occur.

## Discussion
B-cell differentiation into antibody-secreting plasma cells is governed by a complex gene-regulatory network[27,58], epigenetic modifications to chromatin[59–61]and accompanied by large-scale changes in 3D genome organization[31,62,63]. Given the transcriptional dynamics and a large number of cell divisions that B cells undertake it was conceivable that the genome would be progressively remodeled throughout the differentiation process, as has been previously suggested from chromatin accessibility data[61]. However, by performing a detailed spatiotemporal analysis of activation-induced B-cell differentiation, here we reveal that the changes to 3D chromatin structure occur almost exclusively in two waves, prior to the first cell division, which is maintained during subsequent proliferation and clonal expansion, and the second during later differentiation into plasmablasts. Such information not only provides valuable insights into the transcriptional regulation of cellular differentiation but also reveals how immune responses develop and may reveal potential strategies to modulate immunity.

Over the past two decades, a groundswell of evidence has demonstrated that lymphocyte fate decisions are imprinted within a day of antigen exposure, likely prior to the first division[19–24,64–67]. The data presented herein provide a genome-level explanation as to how this could occur. We show that alterations in gene-regulatory chromosome conformation that occur prior to the first cell division and are largely maintained through multiple divisions as the cells clonally expand (see model in Fig. 6f), consistent with high levels of clonal symmetry[18,39]. Our data also suggests that the rapid cell cycle associated with shortened G1

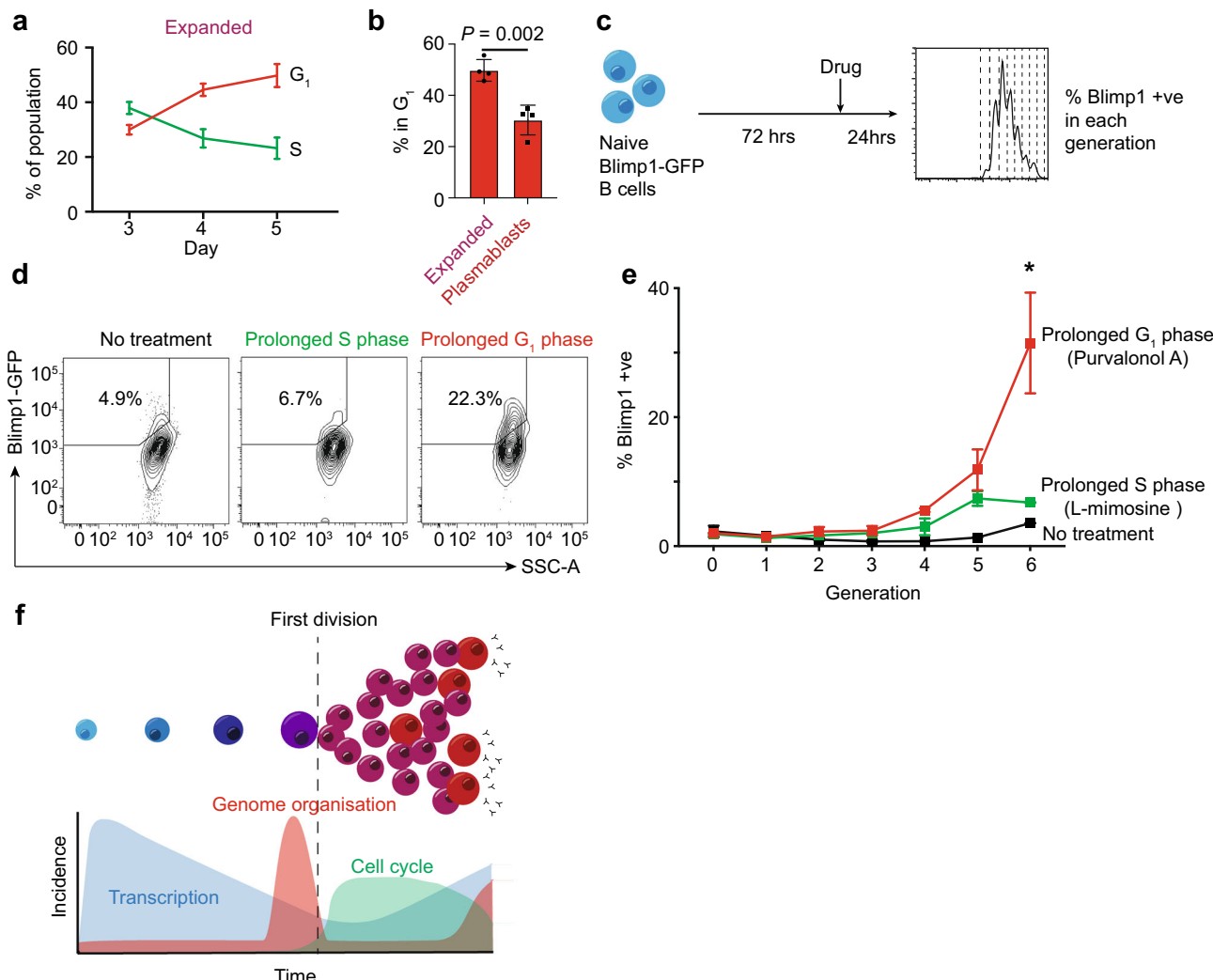

**Fig. 6 Major genome organizational change transitions associate with prolonged time in G₁ phase. a** The percentage of the expanded B-cell population in the $G_1$ or S phase of the cell cycle, as determined by FUCCI, at days 3, 4, and 5 of culture. Mean ± standard deviation from 4 independent experiments is shown. **b** The percentage of the expanded and plasmablast population in the $G_1$ phase of the cell cycle, as determined by FUCCI, at day 5 of culture. The data were using an unpaired two-tailed Student's *t*-test with Welch's correction. Mean ± standard deviation from 4 experiments is shown. **c** Schematic of the experimental set-up for cell cycle manipulations. **d** Flow cytometry plots showing Blimp-GFP B cells cultured with ʟ-mimosine (prolonged S phase), Purvalonal A (prolonged G1 phase) for 24 h or no treatment. Gate and percentage show the percentage of differentiated plasmablasts (Blimp+ve) within generation 6 of the culture. **e** Percentage of plasmablasts (Blimp+ve) within each generation of the Blimp-GFP B-cell 4-day cultures when treated with Purvalonal A (prolonged G1 phase), ʟ-mimosine (prolonged S phase), or no treatment. The data were analyzed using a two-way ANOVA. Mean ± SD from 4 experiments is shown. **f** Model of relationship between transcription, genome organization, and cell cycle. Source data are provided as a Source Data file.

phase might be possible with little genome restructuring, enabling cells to focus on expanding prior to differentiation during the second wave of reorganization that is linked to increased time in G1 (Fig. 6).

Two seminal studies have established when and how the 3D genome is altered during the cell cycle progression of cells undergoing normal turnover[7,8]. These studies observed major alterations in chromosome structure during DNA synthesis and in preparation for mitosis but left open the question of when the genome was restructured during the process of activation-induced differentiation. To address this question, using B-cell activation and differentiation as a model, we revealed a spatio-temporal separation of gene-regulatory chromosome reorganization which occurred in mid-late G1, prior to DNA replication and mitosis. It has previously been considered that DNA synthesis provided an opportunity to remodel gene structure[10,68–70], or that chromosome rewiring to support gene expression during

differentiation would occur after mitosis when chromosomes decondensed in early G1[14]. Overall, we propose that genome reconfiguration is partitioned from the demanding processes of DNA replication and mitosis to ensure the safe implementation of a gene-regulatory program required for the generation of cellular immunity.

## Methods

**Mice**. Experiments were performed on male animals 6–12 weeks of age on a C57BL/6 Pep3b background. Blimp-GFP and FUCCI transgenic mice were described previously[54,71,72]. All mice were maintained at The Walter and Eliza Hall Institute Animal Facility under specific-pathogen-free conditions, ambient temperature between 22–24 °C, humidity 40–60%, and light/dark cycle of 12 h/12 h. All males were randomly chosen from the relevant pool. All experiments were approved by The Walter and Eliza Hall Institute Animal Ethics Committee (#2016-003) and performed under the Australian Code for the care and use of animals for scientific purposes. Results were analyzed without blinding of grouping. Antibodies used in this study can be found in Supplementary Data 13.

**B cells activation and differentiation**. B lymphocytes were cultured in RPMI 1640 with 2 mM GlutaMAX, 50 μM β-mercaptoethanol, and 10% heat-inactivated fetal calf serum (FCS). Splenocytes were obtained from male C57BL/6 mice with red cell lysis and naive resting B cells (TCRβ⁻CD19⁺B220⁺IgM⁺ IgDhi) were sorted on BD FACS Aria III. Naive resting B cells were activated after isolation via negative selection (Miltenyi, 130-090-862). Isolated cells were labeled with 10 μM CellTrace Violet (CTV, Invitrogen), then cultured at a density of $1 \times 10^6$ cells/ml and stimulated with 25 μg/ml lipopolysaccharide (LPS) (Salmonella typhosa origin, Sigma). Activated but undivided cells (TCRβ⁻CD19⁺CD69⁺CTVhighest) were stained and sorted at the corresponding time point. FUCCI B cells were processed as above, except sorted as CD19⁺CD69⁺CTVhighest mAG- mKO2hi at 33 h post-stimulation. For long-term activation, negatively isolated B cells were cultured at a density of $1 \times 10^5$ cells/ml and stimulated with 25 μg/ml LPS. Expanded activated B cells (CD22⁺CD138⁻) and plasmablasts (CD22⁻CD138⁺) were harvested on day 4 post-stimulation. All samples are prepared in biological duplicates for RNA-seq. For in situ HiC, 3, 10, and 33 h post-simulation. Samples are prepared in biological duplicates. Naive resting B cells expanded activated B cells and plasmablasts in situ HiC libraries were used from a previous study GSE99151.

**Manipulating cell cycle**. B lymphocytes were stimulated with 10 μg/ml anti-CD40 (1C10) and 500 units/ml IL-4 for 72 h at $2 \times 10^5$ cells/ml before being incubated with 6 μM Purvalonol A (Sigma) or 150 μM L-mimosine (Sigma) for 24 h.

**In situ HiC**. In situ HiC was performed according to Rao, Huntley[73]. In brief, $7 \times 10^5$–$3 \times 10^6$ cells were resuspended with culture media at $1 \times 10^6$ cells/ml and fixed with 1% v/v formaldehyde (Sigma). Crosslinked cells were lysed with HiC lysis buffer (10 mM Tris-HCl pH8.0, 10 mM NaCl, 0.2% Igepal CA630, and protease inhibitors (Sigma)). Pelleted nuclei were then digested with 100U of MboI (NEB) overnight and subsequently biotin-labeled with Klenow fragment (NEB) and biotin-dATP (Invitrogen). Filled ends were ligated with T4 DNA ligase (NEB) and sonicated (Covaris). The resulting DNA fragments were biotin-pulled down and end-repaired with T4 polynucleotide kinase (NEB), T4 DNA polymerase (NEB) and Klenow fragment, followed by A-tailing with 3′–5′ Klenow (exo-) fragment (NEB), and adaptor ligation using Quick ligase (NEB). The resultant HiC libraries were amplified with Phusion Polymerase (Thermo), size-selected and purified with AMPure XP magnetic beads (Beckman). Purified libraries were sequenced on an Illumina NextSeq 500 to produce 81-bp paired-end reads. Approximately 200 million read pairs were generated per sample.

**HiC pre-processing**. The data pre-processing and analysis were performed as previously described with changes in parameters[34]. In brief, each sample was aligned to the mm10 genome using the diffHic package v1.14.0[37] which utilizes cutadapt v0.9.5[74] and bowtie2 v2.2.5[75] for alignment. The resultant BAM file was sorted by read name, the FixMateInformation command from the Picard suite v1.117 (https://broadinstitute.github.io/picard/) was applied, duplicate reads were marked and then re-sorted by name. Read pairs were determined to be dangling ends and removed if the pairs of inward-facing reads or outward-facing reads on the same chromosome were separated by less than ~1000 bp for inward-facing reads and ~10,000 bp for outward-facing reads. Read pairs with fragment sizes above ~1000 bp were removed. An estimate of alignment error was obtained by comparing the mapping location of the 3′ segment of each chimeric read with that of the 5′ segment of its mate. A mapping error was determined to be present if the two segments were not inward-facing and separated by less than 1000 bp, and around 1–3% were estimated to have errors.

The HOMER HiC pipeline[76–78] was also used for HiC analysis. After processing with the diffHic pipeline, libraries in HDF5 format were converted to the HiC summary format with R. Then input-tag directions were created for each library with the makeTagDirectory function, with the genome (mm10) and restriction-enzyme-recognition site (GATC) specified. Summed biological-replicate tag directories for each cell type were also created.

To determine the reproducibility of the libraries the HiCRep R package was utilized to quantify the similarity between all libraries with the stratum adjusted correlation coefficient (SCC)[79]. For every combination of libraries, the raw contact matrices (50 kbp resolution) of individual chromosomes for each replicate were used to compute the SCC with smoothing parameter 3 and maximum distance considered 5 Mbp. For each pairwise comparison, a median SCC was calculated across all chromosomes.

**Detecting differential interactions (DIs)**. Differential interactions (DIs) between the different libraries were detected using the diffHic package v1.16.0[37]. Read pairs were counted into 50 kbp bin pairs for all autosomes. Bins were discarded if had a count of less than 10, contained blacklisted genomic regions as defined by ENCODE for mm10[80] or were within a centromeric or telomeric region. Filtering of bin-pairs was performed using the filterDirect function, where bin pairs were only retained if they had average interaction intensities more than 4-fold higher than the background ligation frequency. The ligation frequency was estimated from the interchromosomal bin pairs from a 1 Mbp bin-pair count matrix. Diagonal bin pairs were also removed. The counts were normalized between libraries using a loess-based approach with bin pairs less than 1.5 Mbp from the diagonal

normalized separately from other bin pairs. Tests for DIs were performed using the quasi-likelihood (QL) framework[35,36] of the edgeR package v3.26.5[38]. A design matrix was constructed using a one-way layout that specified the cell group to which each library belonged. A mean-dependent trend was fitted to the negative binomial dispersions with the estimateDisp function. A generalized linear model (GLM) was fitted to the counts for each bin pair[81], and the QL dispersion was estimated from the GLM deviance with the glmQLFit function. The QL dispersions were then squeezed toward a second mean-dependent trend, using a robust empirical Bayes strategy[82]. A p-value was computed against the null hypothesis for each bin pair using the QL F-test. p-values were adjusted for multiple testing using the Benjamini–Hochberg method. A DI was defined as a bin pair with a false discovery rate (FDR) below 5%. DIs adjacent in the interaction space was aggregated into clusters using the diClusters function to produce clustered DIs. DIs were merged into a cluster if they overlapped in the interaction space, to a maximum cluster size of 500 kbp. The significance threshold for each bin pair was defined such that the cluster-level FDR was controlled at 5%. Cluster statistics were computed using the csaw package v1.18.0[83]. Overlaps between unclustered bin pairs and genomic intervals were performed using the InteractionSet package[84].

**Detecting TAD boundaries**. TAD breakpoints were detected in each HiC library with TADbit v0.2.0.5 40. Read pairs were counted into 50 kbp bin pairs (with bin boundaries rounded up to the nearest MboI restriction site). The TADbit tool find_tad was run on the raw counts specifying normalized=False. Only TAD boundaries with a score of 7 or higher were included in the results. A one-way ANOVA was used to determine the significance between the median of the TAD sizes at successive stages, after square-root transforming each median.

**Detecting differential TADs**. Differential TAD boundaries (DTB) between B-cell activation stages were detected with the diffHic and edgeR packages[37] using the approach described in Chapter 8 of the diffHic User's Guide. This approach adapts the statistical strategy recently described for differential methylation[85] to identify DTBs. The strength of a putative TAD boundary was assessed based on the upstream vs downstream intensity contrast at that genomic locus, defined as the ratio of upstream to downstream HiC reads anchored at that genomic region. edgeR was used to test whether the ratio at each locus significantly increased or decreased in absolute size between B-cell stages. This method directly assesses differential boundary strength relative to biological variation without needing to make TADs calls in individual samples. Upstream and downstream read counts were determined for 100 kbp genomic regions and for 1 Mbp up- and downstream. Low-abundance regions with average log2-counts per million below 1 were removed. Tests for DTBs were performed using the QL framework of edgeR. Tests were conducted with edgeR glmTreat relative to a fold-change threshold of 1.1 in order to prioritize larger fold-changes[86]. Regions with Treat FDR below 0.05 were considered to be DTB.

**Detection of A/B compartments**. A/B compartments were identified across all the stages of B-cell activation at a resolution of 100 kbp with the method outlined by Lieberman-Aiden et al.[87] with the HOMER HiC pipeline[88]. With the summed biological-replicate tag directories, the runHiCpca.pl function was used on each library with -res 100000 and the genome (mm10) specified to perform principal component analysis to identify compartments. To identify changes in A/B compartments between libraries, the getHiCcorrDiff.pl function was used to directly calculate the difference in correlation profiles.

**Detecting differentially interacting promoters (DIPs)**. Differentially interacting promoters were detected using across all libraries with the diffHic package37. Gene promoters were defined with the genes from the TxDb.Mmusculus.UCSC.mm10. knownGene package v3.4.7 and by applying the promoters function from the GenomicFeatures package v1.36.2 with upstream = 5kbp and downstream = 5kbp. Interactions between the promoters and the entire genome were counted with the connectCounts function from diffHic with a filter set to 0 and the second.region = 10kbp. Interchromosomal interactions were excluded along with interactions contained in blacklisted regions as defined by ENCODE, centromeres or telomeres. Interactions were filtered by abundance as a function of the distance between the anchors. Using the loessFit function from the limma package v3.40.2[89] with span 0.05, a loess curve was fitted to the average log counts per million (logCPM) for all libraries (calculated with the cpmByGroup function of edgeR with log = TRUE) as a function of distance 0.25. An interaction was then required to have an abundance larger than the fitted curved plus two times the mean of the absolute values of the residuals from the loess fit. For each unique promoter, interactions were then aggregated to produce a count matrix. Low-abundance promoters were filtered using edgeR's filterByExpr function with min.count = 200 and min.totals = 200. Obsolete Entrez Gene Ids were removed, as were mitochondrial, ribosomal (rRNA/ncRNA), Riken, olfactory, and X or Y chromosome genes. The counts were normalized between libraries using a loess-based approach with the normOffsets function from the csaw package v1.18.0.

Two different DIPs analyses were performed; with and without the FUCCI B cells HiC libraries. DIPs were detected with the quasi-likelihood (QL) framework[35,36] of the edgeR package. A design matrix was constructed with a

one-way layout that specified the cell type. Using the promoter counts, the estimateDisp function was used to maximize the negative binomial likelihood to estimate the common dispersion across all promoters with trend = none and robust = TRUE[90]. A generalized linear model (GLM) was fitted to the counts[81] and the QL dispersion was estimated from the GLM deviance with the glmQLFit function with robust = TRUE and trend.method = none. The QL dispersions were then squeezed toward a second mean-dependent trend, with a robust empirical Bayes strategy[82] to share information between genes. A p-value was calculated for each promoter using a moderated t-test with glmQLFTest. The Benjamini–Hochberg method was used to control the false discovery rate (FDR) below 5%. Heatmaps of the filtered and normalized logCPM value were plotted with the coolmap function from the limma package. Mean-difference plots (MD plots) were plotted with the plotMD function.

For each DIP in the analysis, the pattern of the promoter interactions was defined across all the transitions as either: 1 (increasing at a transition), −1 (decreasing at a transition) or 0 (not significantly changed). An individual DIP was determined to be in a pattern if it had FDR below 5% at that transition while the sign of the logFC gave the direction.

**Motif enrichment in DIPs.** A stricter DIPs analysis was performed as described in the section above without the FUCCI B cells HiC libraries. In this analysis for an interaction to be included, it was required to have an abundance larger than the fitted curved plus three times the mean of the absolute values of the residuals from the loess fit. DIPs were again grouped into patterns and all the interactions contributing to a pattern were defined. The getfasta function from Bedtools was used to obtain the fasta sequence from all the non-promoter anchors for interactions in a pattern. Relative motif enrichment was performed with the program ame of the MEME v5.0.5[91] software package on each pattern. The motif database used was HOCOMOCOv11_full_MOUSE from MEME. Enriched motifs were required to have p-value < 0.05, adjusted for multiple tests using a Bonferroni correction. The -control flag was used with a set of control sequences as the background. The control sequences were the non-promoter anchors of 1500 randomly selected non-DIPs across B-cell activation. Motifs were removed if the corresponding protein was not expressed (RPKM < 1).

**Detecting looping interactions.** Looping interactions were detected in the naive B-cell libraries with a method similar to that described by Rao et al.[73] and as described previously[34] except with a bin size of 20 kbp. Established loops were defined as those with enrichment values above 0.5, with an average count across libraries >5, and that were more than 60 kbp away from the diagonal. Loops were annotated with genes in the anchors with annotatePairs function from the diffHic package. The significance of the enrichment of DE genes between naive B cells and 3 h after activation (without logFC threshold) in the loops was performed with a Pearson's chi-squared test (df = 2). The test was between the portions of genes within/not within loop anchors that are: DE genes increasing, DE genes decreasing, or non-DE genes.

**Visualization of HiC.** Multidimensional-scaling plots were constructed with the plotMDS function in the limma package applied to the filtered and normalized logCPM values of each bin pair or promoter for each library. The distance between each pair of samples was the 'leading log fold change', defined as the root-mean-square average of the 500 largest log2-fold changes between that pair of samples.

Normalized contact matrices at a 50 kbp resolution were produced with HOMER HiC pipeline for visualization. With the summed biological-replicate tag directories, the analyzeHiC function was used with the -balance option. This implements the matrix balancing approach which iteratively balances matrices to ensure that the total interactions are the same for each column and row[92] Plaid plots were constructed using the plotHic function from the Sushi R package v1.22.0[93]. The color palette was inferno from the viridisLite package v0.3.0[94]. The plotBedpe function of the Sushi package was used to plot the interactions contributing to DIPs.

**RNA-seq.** RNA was extracted with NucleoSpin RNA Plus (Macherey-Nagel) and subsequently quantified in a TapeStation 2200 using RNA ScreenTape (Agilent). Libraries were prepared with a TruSeq RNA sample-preparation kit (Illumina) from 500 ng RNA, as per the manufacturers' instructions. Libraries were then amplified with KAPA HiFi HotStart ReadyMix (Kapa Biosystems) and 200 to 400 bp products were size-selected and cleaned up with AMPure XP magnetic beads (Beckman). Final libraries were quantified with TapeStation 2200 using D1000 ScreenTape for sequencing on the Illumina NextSeq 500 platform to produce 81 bp paired-end reads. Around 25 million read pairs were generated per sample.

Reads were aligned to the mm10 genome with Rsubread package v1.28.0 Genewise counts were obtained with featureCounts and Rsubread's inbuilt Entrez gene annotation. Low-abundance genes were filtered out with the filterByExpr function. Obsolete Entrez Gene Ids were removed, as were mitochondrial, ribosomal, X or Y chromosome genes and variable immunoglobulin gene segments. Normalization was performed with the trimmed mean of M-values (TMM) method[95].

The differential expression analysis was conducted using the quasi-likelihood (QL) framework[35,36] of edgeR. The empirical Bayes procedure of glmQLFTest was run in robust mode to increase power and protect against hypervariable genes[82]. Differential expression was evaluated using edgeR's glmTreat function with the log-fold-change threshold set to 1.5[86]. This method prioritizes genes with biological significant changes by ensuring that all differentially expressed genes have observed fold-changes significantly >1.5. The Benjamini–Hochberg method was used to control the FDR below 5%. Log2 RPKM (reads per kilobase per million) were computed with edgeR's rpkm function. MDS plots, MD plots, and heatmaps were created using the plotMDS, plotMD, and coolmap functions. Distances on the MDS plots correspond to leading log2-fold-changes from the top 500 differentially expressed genes.

**Patterns of differential expression were determined for genes with the same method as the DIPs**

*Gene set enrichment.* Tests for over-representation of gene ontology terms were performed with the goana function from the limma package. Gene set enrichment was tested using limma's fry function and visualized using limma's barcode enrichment plot. Terms containing less than 100 genes were removed. Categories were ranked according to the percentage of genes within the category that are differentially expressed. Terms in the "CC" ontology category were removed.

*Associating genomic regions with ChIP-seq profiling.* The raw read sequence data of ChIP-seq profiles of Irf4 binding sites in lipopolysaccharide activated B cells and plasmablasts and Blimp1 binding sites in plasmablasts were downloaded from the Gene Expression Omnibus (GEO) repository GSE71698 46. Raw reads were aligned to the mm10 genome with RSubread and duplicate reads were removed with MarkDuplicates. Peaks were called with MACS2 v2.1.1[96] with default parameters. Overlaps between all the non-promoter anchors for interactions in a pattern and called peaks were identified with the overlapsAny function of the IRanges package v2.20.2[97]. Pearson's chi-square test with Yates' continuity correction (degrees of freedom = 1) was used to determine the significance of differences in overlapping proportions.

**Reporting summary.** Further information on research design is available in the Nature Research Reporting Summary linked to this article.

## Data availability

Sequencing data that support the findings of this study are tabulated in the supplementary tables and are available in the GEO database under accession numbers "GSE147497" and "GSE99151". All other relevant data supporting the key findings of this study are available within the article and its Supplementary Information files or from the corresponding author upon reasonable request. A reporting summary for this Article is available as a Supplementary Information file. Source data are provided with this paper.

## Code availability

DiffHic and edgeR, which are freely available from the Bioconductor repository (https://bioconductor.org/packages/release/bioc/html/diffHic.html and https://bioconductor.org/packages/release/bioc/html/edgeR.html, respectively) were used in this study. Versions used for this manuscript were: diffHic v1.16.0[37], bowtie2 v2.2.5[75], Picard suite v1.117 (https://broadinstitute.github.io/picard/), quasi-likelihood (QL) framework[35] of the edgeR package v3.26.5, csaw package v1.18.0[83], TADbit v0.2.0.5 python-based software[98], limma package v3.40.2[89], Sushi R package v1.22.0[99], rtracklayer package v1.32.2[100], Rsubread v1.28.0[101], MACS2 v2.1.0[96], HOMER v4.11[76], MEME v5.0.5[91], cutadapt v0.9.5[102], HiCRep v1.14.0[79], viridisLite v0.3.0[94], and IRanges v2.20.2[97].

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

## Acknowledgements

We thank the staff of the core facilities at the Walter and Eliza Hall Institute. This work was supported by grants and fellowships from the H.C. Marian and E.H. Flack Fellowship (H.D.C.), the National Health and Medical Research Council of Australia (C.R.K #1125436, P.D.H. #1054925, T.M.J #1124081, R.S.A #1100451, G.K.S & R.S.A #1158531), and the Australian Research Council (R.S.A. #130100541). This study was made possible through Victorian State Government Operational Infrastructure Support, the Australian Government NHMRC Independent Research Institute Infrastructure Support scheme, and the Australian Cancer Research Fund.

## Author contributions

W.F.C., J.H.S.Z., C.R.K., N.G.B., and T.M.J. designed and conducted experiments. H.D.C. and G.K.S. performed computational analyses. P.D.H., T.M.J., and R.S.A. conceived the study and wrote the manuscript with assistance from H.D.C and G.K.S.

## Competing interests

The authors declare no competing interests.
