## [Peer Review File · Nature Communications]

REVIEWER COMMENTS

Reviewer #1 (Remarks to the Author):

Pre-mitotic genome organization establishes the transcriptional imprint for cellular differentiation

In this paper, Chan and Coughlan et al. reveal two waves of chromosome conformational changes that occur during developmental progression from the naïve B to the plasma-blast cell stage. The data indicate that alterations in 3D chromatin structure occur in two discrete windows in G1. Based on these observations, the authors conclude that chromosome reconfiguration is spatiotemporally separated from DNA replication and mitosis to instruct a regulatory circuitry that orchestrates plasma cell fate.

This is an interesting study. The authors provide intriguing insights into how nuclear architecture and cell cycle progression are linked to establish plasma cell identity. Overall the data are compelling and of interest. There are a few issues that need to be addressed to further improve the manuscript.

Major Comments:

1. Could the authors please include a table indicating genomic regions that change PC1 values for the various developmental stages that were examined. Minor changes should be indicated since we are dealing here with purified yet mixed/heterogeneous populations. Such changes, albeit minor, should then be compared to differences in transcription.
2. Could the authors please include CTCF and RAD21/SMC1/SMC3 ChIP-Seq data for the developmental stages that were examined in the study. Changes in CTCF and RAD21 occupancy should then be compared to alterations in looping.
3. It would be of interest to determine whether and how changes in Prdm1 occupancy during the final stages of plasma cell differentiation relate to alterations in long-range genomic interactions.

Minor Comment:

The authors need to check for typos and grammatical errors. Figure Legends lack periods and are full of typos and errors.

Reviewer #2 (Remarks to the Author):

In this manuscript, the authors perform HiC and RNA-seq on B cells at varying stages of B cell activation and conclude transcriptional changes are concurrent with genome organisational changes. The paper is premised upon the comparability of the HiC results between cells at 6 B cell activate stages.

The basic statistics regarding these HiC results are not clear. For instance, the number of interaction pairs, distance decay curves, whether or not there are replicates, and then the reproducibility scores, etc. In Fig1f the colour bars for all the HiC maps have different ranges, ideally they should all be the same range with the same lower limit and upper limit to make the maps comparable to one another visually. The paper lacks statistics throughout; a lot of the comparisons don't have associated p values.

Main Points.

1) Overall I found the paper difficult to read, often times there are claims that I can't find evidence to substantiate. The one sentence summary, i.e. the DNA replication and mitosis independent genome organisation 'controlling' cell differentiation is not supported by the evidence presented. This is a descriptive study - there is no causal dissection of genome organisational changes and transcriptional changes so such conclusions are unwarranted. At a minimum, to begin to make such statements, the authors could have performed HiC on the drug treated cells used in Fig6.

2) The figure legend to Fig2 reads 'Large genomic structures are stable, except for changes just prior to mitosis'. I don't think that claim is supported by the figure itself. For instance, naïve B cells have smaller TADs and more short range counts. There is no p value so the statistical significance

of this is unknown but there are clearly changes from naïve B cells to the next cell stage that are unacknowledged by the authors. Moreover, the distributions of DI spans of differential domains between expanded and plasmablasts are also different.

3) The figure legend to Fig4 reads 'Numerous transcription factors regulate B cell differentiation via three dimensional genome organisation'. We cannot conclude causally – these statements should be modified. The binding of transcription factors at DIPs doesn't necessarily mean these factors are implicated in effectuating genome organisational changes. Moreover, the appearance of motifs does not necessarily guarantee binding. Finally, from Fig4g and h it is hard to see that the binding motifs of transcription factors are consistent with their supposed roles in B cell activation as there are so many labelled genes, whose functions might not be apparent to the readers. I wonder if it would be better to analyse a few specific transcription factors whose binding sites are known (via ChIPseq for instance) and plot the interaction differences as a function of gene expression changes.

4) The authors show only genomics data so these are correlative experiments from which causal conclusions cannot be drawn. The authors should comb the paper to remove statements such as the following: "This suggests that the dominant function of promoter-enhancer interaction during B cell differentiation is to drive expression" – there is no basis for this suggestion there is only a correlation from which it is equally possible that transcription drives structure or that a third correlated feature not measured is the driver. Here is another: "suggests that many more transcription factors than previously demonstrated regulate immune cell gene expression via a three-dimensional genome organisation."

5) One attempt at a causality experiment by treating with drugs Puvalonol A and mimosine. These drugs are poorly characterized – the Puvalonol reference is very old. Mimosine is still controversial and its effects are known to be cell type specific. I would be very careful drawing conclusions from these experiments.

Minor points:

1) Under data accessibility the second GSE number is marked with stars instead of actual number.

2) Fig3d, what is the y axis, i.e. what's the numerical scale of up and down regulated interactions?

3) Fig3e can be accompanied by a similar heatmap but showing all the DIPs. How many of DIPs show this 'two waves' of changes? Can Fig3f be expanded to include all DIPs, not just the several selected key b cell genes?

4) If you add up all the genes they claim are significantly altered, it amounts to nearly all the annotated genes in the genome. Does this include housekeeping genes? Can the authors explain? Are some going up and then down again perhaps?

5) The authors reference classical papers for chromosome re-positioning during early G1 phase going back to 2003, but are missing the original paper: Dimitrova et. al., Molecular Cell, 1999.

6) Line 12 and 17 – Imminently I think should read immediately. And "imminent division population" – what does that mean?

Reviewer #3 (Remarks to the Author):

Pre-mitotic genome reorganization establishes the transcriptional imprint for cellular differentiation

Here the authors perform a detailed spatiotemporal analysis of changes in genome structure during AID-induced B cell differentiation and show that changes occur in mid-late G1, a stage that they demonstrate is important for promoting differentiation. These alterations in chromosome structure are then largely maintained through cell division after differentiation. While this conclusions may be solid there are numerous other unsupported conclusions and issues that detract from the paper.

The paper is difficult to read and the data hard to interpret because a lot of information is missing in the main text about how the data were analyzed and this should be included so the reader can follow what has been done rather than having to search in the methods for a detailed explanation.

There are many statements that are unsupported by the data. For example, there is the

assumption that all promoter interactions are made with enhancers, however there is no data to support that targets are either enhancers or putative enhancers. These type of unsupportive statements are made again and again.

There is an assumption made that gene expression is regulated by 3-D genome organization but there is no evidence to support this – it could equally well be the other way around and in fact this is a hotly debated topic in the field. Statements of this type are made throughout the paper.

Another example is the following statement: 'Irf4 is indeed regulating gene expression via three-dimensional genome organization'. There is no data to support this claim – such as how many genes with changes in interactions bind or don't bind Irf4. Even if this data was provided the evidence would be purely correlative and it would be necessary to delete Irf4 or better still, mutate some of the Irf4 binding sites to substantiate the claim.

Changes in gene activity linked to changes in interaction is not a new concept. Also well documented is the involvement of the TFs at the different stages of B cell development.

I fail to see the logic behind the statement that increased expression of a TF is associated with its function as a gene activator while decreased expression is associated with its function as a repressor. How then do they explain the fact that numerous TFs like Pax5 and Bcl6 are both repressors and activators?

There is potential novelty and interest in the investigation of TFA and its links to genome organization but this is not explored sufficiently to draw any strong conclusions.

The authors state that it is currently unclear if genome reorganization occurs throughout interphase or at a specific stage of the cell cycle. This has already been explored – see studies from Peter Fraser's and Job Dekker's labs for example. The findings from the current study should be discussed in light of previous findings.

Specific comments.

Figure 1 – The resolution is too low to identify interactions between the Twistnb promoters and any 'enhancers'. What are the authors referring to as enhancers – validated or putative enhancers?

When the authors talk about structural changes around the Bcl6 gene – they should put these in the context of what was found by the Melnick lab (Bunting et al., 2017).

Fig 1I. The authors claim that the transcriptional changes that occur just prior to the first cell division are almost exclusively involved in chromatin remodeling and DNA conformation – however this is not an accurate description of what is shown in the figure which has more to do with replication, mitosis and cell cycle as would be expected.

Figure 3A – there is no scale on the x axis. Also there is no explanation of how they distinguish between strong and weak interactions.

Figure 6D – the name of the drug should be included in the figure.

Page 9 line 20: from the text it is not clear where motifs were analyzed: at the promoters from the list of DIP, or at the suspected enhancers interacting with those promoters?

REVIEWER COMMENTS

Reviewer #1 (Remarks to the Author):

This is an interesting study. The authors provide intriguing insights into how nuclear architecture and cell cycle progression are linked to establish plasma cell identity. Overall the data are compelling and of interest. There are a few issues that need to be addressed to further improve the manuscript.

Major Comments:

1. Could the authors please include a table indicating genomic regions that change PC1 values for the various developmental stages that were examined. Minor changes should be indicated since we are dealing here with purified yet mixed/heterogeneous populations. Such changes, albeit minor, should then be compared to differences in transcription.

As requested, the A/B compartments that change state at each transition of B cell differentiation have been tabulated and included in the revised manuscript as Supp Table 6. As originally stated in the manuscript the changes are minimal and no correlation to gene expression changes could be detected.

2. Could the authors please include CTCF and RAD21/SMC1/SMC3 ChIP-Seq data for the developmental stages that were examined in the study. Changes in CTCF and RAD21 occupancy should then be compared to alterations in looping.

We thank the Reviewer for their suggestion. Overlaying publicly available ChIP data for CTCF and Rad21 in naïve B cells and imminently dividing B cells with our DIs between these stages we find a significant enrichment (both $P < 2.2e-16$) of CTCF and Rad21 binding within the anchors of DIs, compared to unchanged genome organisation. While these results are significant and build confidence in our data, we feel that they add little to, and possibly distract from, the narrative of the manuscript and request that they not be included in the revised manuscript.

3. It would be of interest to determine whether and how changes in Prdm1 occupancy during the final stages of plasma cell differentiation relate to alterations in long-range genomic interactions.

We thank the Reviewer for their interest and agree that the role of Blimp1/*Prdm1* in potentially regulating three-dimensional genome organization of plasmablasts is very interesting and is, in fact, predicted by our motif analysis. By overlaying ChIP data for Blimp1 binding with our genome organizational data, we show that “Blimp1 binding is significantly enriched ($P = 3.83 \times 10^{-35}$) in the putative long-range enhancers of genes with decreased interactivity, compared to those with increased interactivity.”

Minor Comment:

The authors need to check for typos and grammatical errors. Figure Legends lack periods and are full of typos and errors.

We thank the Reviewer for their attention to detail. The manuscript has been extensively revised to ensure accuracy of language in response to this concern and the comments of

other Reviewers.

Reviewer #2 (Remarks to the Author):

In this manuscript, the authors perform HiC and RNA-seq on B cells at varying stages of B cell activation and conclude transcriptional changes are concurrent with genome organisational changes. The paper is premised upon the comparability of the HiC results between cells at 6 B cell activate stages.

The basic statistics regarding these HiC results are not clear. For instance, the number of interaction pairs, distance decay curves, whether or not there are replicates, and then the reproducibility scores, etc.

We thank the Reviewer for their attention to detail. We have included a new Supp Table (Supp Table 1) within the revised manuscript containing the requested library statistics for all *in situ* HiC libraries.

In Fig1f the colour bars for all the HiC maps have different ranges, ideally they should all be the same range with the same lower limit and upper limit to make the maps comparable to one another visually.

We thank the Reviewer for their attention to detail. As the Reviewer rightly states the scale bars of the contact matrices in Fig 1 F are all different, however, the reason for this is that the libraries are (understandably) all of slightly different read depths and have been scaled so that the matrices are visually comparable. This scaling is a standard in the field.

The paper lacks statistics throughout; a lot of the comparisons don't have associated p values.

We thank the Reviewer for their constructive comment and note this was also a concern of other Reviewers. While the vast majority of the conclusions in the manuscript were based on statistical significance, we have now revised the manuscript to state the associated P values.

Main Points.

- 1) Overall I found the paper difficult to read, often times there are claims that I can't find evidence to substantiate. The one sentence summary, i.e. the DNA replication and mitosis independent genome organisation 'controlling' cell differentiation is not supported by the evidence presented. This is a descriptive study - there is no causal dissection of genome organisational changes and transcriptional changes so such conclusions are unwarranted. At a minimum, to begin to make such statements, the authors could have performed HiC on the drug treated cells used in Fig6.

We thank Reviewer 2 for their constructive critique. We agree that the work is descriptive and it appears that in places we have overdrawn our conclusions. As a point of clarification, the statement regarding genome organisation "controlling" differentiation by the Reviewer is a misquote. The sentence in question read: "Overall, we propose that chromosome reconfiguration is spatiotemporally separated from DNA replication and mitosis to ensure the implementation of a gene regulatory program that controls the differentiation process required for the generation of immunity.". This sentence implies not that genome

organisation controls differentiation, but the gene regulatory program. Nonetheless, significant changes have been made to the revised manuscript to more fairly and accurately reflect our findings.

- 2) The figure legend to Fig2 reads 'Large genomic structures are stable, except for changes just prior to mitosis'. I don't think that claim is supported by the figure itself. For instance, naïve B cells have smaller TADs and more short range counts. There is no p value so the statistical significance of this is unknown but there are clearly changes from naïve B cells to the next cell stage that are unacknowledged by the authors. Moreover, the distributions of DI spans of differential domains between expanded and plasmablasts (Fig 2f) are also different.

We thank the Reviewer for their attention to detail. In response to this concern we have reanalyzed TAD sizes across all samples with each replicate separated, as opposed to pooled. This allowed P value determination. As rightly concluded by the Reviewer significant changes do occur in the early stages of B cell activation as well as prior to mitosis. Figure 2 and the manuscript have been revised to outline these new analyses and conclusions.

- 3) The figure legend to Fig4 reads 'Numerous transcription factors regulate B cell differentiation via three dimensional genome organisation'. We cannot conclude causally – these statements should be modified. The binding of transcription factors at DIPs doesn't necessarily mean these factors are implicated in effectuating genome organisational changes. Moreover, the appearance of motifs does not necessarily guarantee binding.

We acknowledge that the title was an overstatement of our findings. It has been modified in the revised manuscript to read "Numerous transcription factor motifs associate with three-dimensional genome organizational changes during B cell differentiation". The associated paragraph has also been extensively modified to more accurately reflect our findings.

...Finally, from Fig4g and h it is hard to see that the binding motifs of transcription factors are consistent with their supposed roles in B cell activation as there are so many labelled genes, whose functions might not be apparent to the readers. I wonder if it would be better to analyse a few specific transcription factors whose binding sites are known (via ChIPseq for instance) and plot the interaction differences as a function of gene expression changes.

We thank the reviewer for their comments. We agree that Fig 4 G and H are visually busy. Regarding the examination of individual transcription factors and their role in organizing the genome we refer the Reviewer to our findings using Irf4 and Blimp1 ChiP-Seq (end of the transcription factor motif section, p11).

- 4) The authors show only genomics data so these are correlative experiments from which causal conclusions cannot be drawn. The authors should comb the paper to remove statements such as the following: "This suggests that the dominant function of promoter-enhancer interaction during B cell differentiation is to drive expression" – there is no basis for this suggestion there is only a correlation from which it is equally possible that transcription drives structure or that a third correlated feature not measured is the driver. Here is another: "suggests that many more transcription factors than previously demonstrated regulate immune cell gene expression vi 5 a three-dimensional genome

organisation.”

Again, we acknowledge that the work is purely descriptive. In response to Reviewers comments we have removed or re-worded many overdrawn statements within the revised manuscript. For example, “long-range enhancers” are now “putative long-range enhancers”, among others.

5) One attempt at a causality experiment by treating with drugs Puvalonol A and mimosine. These drugs are poorly characterized – the Puvalonol reference is very old. Mimosine is still controversial and its effects are known to be cell type specific. I would be very careful drawing conclusions from these experiments.

We thank the Reviewer for their cautionary comment. We feel that the language within this section of the manuscript reflects these weaknesses and is appropriately cautious.

Minor points:

- 1) Under data accessibility the second GSE number is marked with stars instead of actual number.

The appropriate GSE has been inserted into the revised manuscript.

2) what is the y axis, i.e. what’s the numerical scale of up and down regulated interactions?

We thank the Reviewer for the clarification request. The y axis in Fig 3 D is the change of state for each DIP with only three options: up, down or no change. This change of state is determined by the differential analysis of each DIP at each transition. Thus, the plot shows the trajectory of all DIPs across B cell differentiation.

3) Fig3e can be accompanied by a similar heatmap but showing all the DIPs. How many of DIPs show this ‘two waves’ of changes? Can Fig3f be expanded to include all DIPs, not just the several selected key b cell genes?

We refer the Reviewer to Supplemental Figure 2 F for heatmaps of all DIPs at each transition.

- 4) If you add up all the genes they claim are significantly altered, it amounts to nearly all the annotated genes in the genome. Doe this include housekeeping genes? Can the authors explain? Are some going up and then down again perhaps?

If we sum all the DEs detected across all transitions of B cell differentiation we find 12,326 significant transcriptional changes during B cell differentiation. We also detect >17,000 significant changes in genome organization (DIs). Both DEs and DIs include “housekeeping genes” and both include increases and decreases at each transition (see Fig 1 C and E). Changes in expression or structure can be enhanced or reversed at any other transition. Each transition is analysed independently of all others.

5) The authors reference classical papers for chromosome re-positioning during early G1 phase going back to 2003, but are missing the original paper: Dimitrova et. al., Molecular Cell, 1999.

We thank the Reviewer for their attention to detail and apologise for the oversight. This important reference has been added to the revised manuscript.

6) Line 12 and 17 – Imminently I think should read immediately. And “imminent division population” – what does that mean?

We thank the Reviewer for their proposal. ‘Imminent division’ is meant to imply that the population is just about to divide. ‘Immediately’ would imply a similar state, but has more out of context meanings which we chose to avoid. We are of course willing to change if deemed necessary by the Editor or Reviewer.

Reviewer #3 (Remarks to the Author):

Here the authors perform a detailed spatiotemporal analysis of changes in genome structure during AID-induced B cell differentiation and show that changes occur in mid-late G1, a stage that they demonstrate is important for promoting differentiation. These alterations in chromosome structure are then largely maintained through cell division after differentiation. While this conclusions may be solid there are numerous other unsupported conclusions and issues that detract from the paper.

The paper is difficult to read and the data hard to interpret because a lot of information is missing in the main text about how the data were analyzed and this should be included so the reader can follow what has been done rather than having to search in the methods for a detailed explanation.

We thank the Reviewer for their constructive comment. The manuscript has been extensively revised to include detailed descriptions of statistical procedures, including RNA-Seq analysis, *in situ* HiC analysis, and TAD boundary calling.

There are many statements that are unsupported by the data. For example, there is the assumption that all promoter interactions are made with enhancers, however there is no data to support that targets are either enhancers or putative enhancers. These type of unsupportive statements are made again and again.

We thank the Reviewer for their valid criticism, and note that this was a consistent concern across Reviewers. In response we have made extensive changes to our conclusionary statements throughout the revised manuscript to ensure they are no longer overdrawn.

There is an assumption made that gene expression is regulated by 3-D genome organization but there is no evidence to support this – it could equally well be the other way around and in fact this is a hotly debated topic in the field. Statements of this type are made throughout the paper.

Again, we thank the Reviewer for their comments and, as above, refer to the extensive changes made to the revised manuscript endeavoring to make our conclusionary comments fair and accurate.

Another example is the following statement: ‘Irf4 is indeed regulating gene expression via three-dimensional genome organization’. There is no data to support this claim – such as how many genes with changes in interactions bind or don’t bind Irf4. Even if this data was

provided the evidence would be purely correlative and it would be necessary to delete Irf4 or better still, mutate some of the Irf4 binding sites to substantiate the claim.

We acknowledge both that the work is purely descriptive and that the conclusion referred to by the Reviewer was overdrawn. We apologise. The offending statement has been heavily revised.

Changes in gene activity linked to changes in interaction is not a new concept. Also well documented is the involvement of the TFs at the different stages of B cell development. I fail to see the logic behind the statement that increased expression of a TF is associated with its function as a gene activator while decreased expression is associated with its function as a repressor. How then do they explain the fact that numerous TFs like Pax5 and Bcl6 are both repressors and activators? There is potential novelty and interest in the investigation of TFA and its links to genome organization but this is not explored sufficiently to draw any strong conclusions.

We thank the Reviewer for their comment. To clarify, our analyses simply highlight transcription factors with motifs that are significantly enriched in the putative long-range enhancers of genes that significantly change interactivity during B cell differentiation transitions. This can be an increase or decrease in interactivity. We then speculate that if the expression of these transcription factors also changes at that transition then perhaps these changes are linked. Obviously, this is associative and cannot dissect the subtleties of transcription factor function or binding.

The authors state that it is currently unclear if genome reorganization occurs throughout interphase or at a specific stage of the cell cycle. This has already been explored – see studies from Peter Fraser’s and Job Dekker’s labs for example. The findings from the current study should be discussed in light of previous findings.

The statement on page 12 was in reference to our current study and has been altered to reflect this. However, neither Peter Fraser’s nor Job Dekker’s works examined how the genome was reorganized in response to differentiation signals as we have done here. We have now referenced the Nagano et al., Nature 2017 in the discussion to better reflect previous works.

Specific comments.

Figure 1 – The resolution is too low to identify interactions between the Twistnb promoters and any ‘enhancers’. What are the authors referring to as enhancers – validated or putative enhancers?

We thank the Reviewer for their attention to detail. In response to their concern, and others, the manuscript has been extensively revised, including to clarify that ‘enhancers’ identified by HiC are “..putative long-range enhancers...”.

When the authors talk about structural changes around the Bcl6 gene – they should put these in the context of what was found by the Melnick lab (Bunting et al., 2017).

We thank the reviewer for their attention to detail and apologise for the oversight. This important reference has been added to the revised manuscript.

Fig 1I. The authors claim that the transcriptional changes that occur just prior to the first cell division are almost exclusively involved in chromatin remodeling and DNA conformation – however this is not an accurate description of what is shown in the figure which has more to do with replication, mitosis and cell cycle as would be expected.

We thank the Reviewer for their attention to detail and apologise for the inaccuracy of language. The sentence in question now reads: “However, interestingly, the predicted function of the transcriptional changes in the hours just prior to the first cell division **include** chromatin remodeling and DNA conformation change (Fig 1 I)(Supp Table 4).”

Figure 3A – there is no scale on the x axis. Also there is no explanation of how they distinguish between strong and weak interactions.

We thank the Reviewer for their attention to detail. Regarding the x axis label of Fig 3 A please find the axis label below the associated contact matrices within the same figure panel. It is simply a linear genomic scale. If desired it can be moved or duplicated, however, we feel its position highlights the fact that the two plot types show the same data and should be viewed together.

Regarding ‘strong’ and ‘weak’ interactions, there is no distinction. The red and green lined interactions are intended to be simply representative, and were selected as they had the highest CPM and linear span. All DIPs are plotted.

Figure 6D – the name of the drug should be included in the figure.

The names of the drugs used has been added to a revised Figure 6.

Page 9 line 20: from the text it is not clear where motifs were analyzed: at the promoters from the list of DIP, or at the suspected enhancers interacting with those promoters?

We thank the Reviewer for their request. The revised section in question now reads: “To explore what may be regulating this three-dimensional gene regulatory network we determine the prevalence on DNA binding motifs of transcription factors (>1 RPKM) within the putative long-range enhancers of DIPs, relative to long-range enhancers that are not differential during B cell differentiation.”

Reviewers' comments:

Reviewer #1 (Remarks to the Author):

The manuscript has improved. In my opinion it is nice work and recommend publication in Nature Communication. One minor detail. A recent manuscript describes changes in compartmentalization during plasma cell differentiation (Bortnick et al., 2020). I think this paper should be referenced.

Reviewer #2 (Remarks to the Author):

In this revised manuscript the authors have done surprisingly little to address what were substantive criticisms from the reviewers. The most offensive of all was the causality statement in the title of the paper, which suggests causality and an order of events that is completely unsubstantiated by the data. This is unacceptable.

Regarding my overall comments:

In response to Point1, the authors did not provide distance decay curves or reproducibility scores. These are not details as the authors claim, but are essential to interpretation of the data.

The authors chose to ignore our Point2, stating that their display of the data "is a standard in the field", which is simply not true. There are several ways to normalise matrices so that they are comparable between samples, for instance iterative correction normalisation or even show the $\log_2(\text{observed over expected})$. Since the authors are comparing between cell stages, this is quite important.

Main points:

Point 1 My criticism, shared by all reviewers and strongly by reviewer #3, was that there is no causal dissection of genome organisational changes and transcriptional changes so such conclusions are unwarranted'. We suggested an experiment that could permit preliminary causal conclusions, which they did not respond to. They claim to have made 'extensive changes' in wording but I could only find small changes that did not revise their interpretation of the data in a way that is consistent with the descriptive nature of the data.

Point 2 The authors performed the analysis we asked for and changed the figure legend accordingly. However, the original manuscript was largely predicated on the fact that changes in architecture only occur prior to mitosis when transcription changes for clonal expansion are established. The results of this new analysis of the same data challenge that conclusion. But the authors do not incorporate this result into their overall conclusions. The authors should at least integrate this new result in their later analysis, showing the fraction of changes that have no gene transcription consequences, etc.

Point3 The authors say they made changes but the statements we pointed out remain completely unchanged and they did not address any of our criticisms of the figure itself with any revisions.

Point4-5, same issue, adding 'putative' to a phrase does not equate with the careful reassessment of the relationship between form and function.

Minor points:

Point3 We asked for Fig3f to be expanded to include all DIPs not just key genes, and this was ignored.

Point4 The authors did not indicate any changes in the manuscript in response to this point. There

are indeed a lot of genes that changed expression, more than half of all genes, some of them housekeeping genes... Are they all or mostly accompanied by some sort of interaction changes? If so, the genome organisational changes could very well be a reflection of the changes in transcriptional output instead of the cause of transcription changes specific to differentiation.

Reviewer #3 (Remarks to the Author):

Pre-mitotic genome reorganization establishes the transcriptional imprint for cellular differentiation

I was disappointed to find that very little was changed during the revision and the authors continue to make statements that are not supported by their data. In addition, they fail to adequately discuss their work in the light of papers from the Fraser and Dekker labs who have both done extensive analyses regarding changes in chromatin organization during cell cycle / cell division.

Abstract: These changes reveal an elaborate gene-regulatory network providing an explanation for how lymphocyte fate is imprinted prior to the first division'

Is an overstatement as the authors did not generate a regulatory network. I suggest the authors read up on regulatory network inference before making claims of this sort.

Line 17 – the authors still have not referenced the work of Dekker and Fraser

Line 25: Overall, we propose that chromosome reconfiguration is spatiotemporally separated from DNA replication and

Mitosis – this not knew and has been shown previously by Dekker and Fraser

Page 6 Line 12: As such, strong three-dimensional connections between the Twistnb promoter and putative long-range enhancers are detected prior to the first activation-induced division but not after (Fig 1 F). There is no evidence for these being putative enhancers – if the authors want to make this claim they should overlap with ATAC-seq and/or H3K27ac data at the same time points to show this.

Our organizational data suggests many of the changes in genome organization we observe are likely promoter-enhancer interactions.

Page 10: 'hinting that many more transcription factors than previously demonstrated may regulate immune cell gene expression via three-dimensional genome organisation (Hu et al., 2018; Johanson et al., 2018a).

This sentence should be changed to 'hinting that many more transcription factors than previously demonstrated may regulate immune cell gene expression and these changes are linked to changes in three-dimensional genome organisation'.

This way there is no cause or effect implied.

Page 12: Thus, consistent with our association analyses and those of others, the presence of Irf4 or Blimp1 within promoter interacting regions significantly increases the chance that these structures are removed or repressed.

This sentence should be changed to 'Thus, consistent with our association analyses and those of others, the presence of Irf4 or Blimp1 within promoter interacting regions significantly increases the chance that these structures are altered.

Page 12: Taken together these analyses suggest that overlays of transcription factor expression data and genome organizational data may be used to infer the intricate and previously undetectable regulatory network of TFs influencing B cell activation and differentiation.

The authors have linked changes in TF expression to changes in chromatin structure – they have definitely NOT inferred a regulatory network of TFs influencing B cell activation. B cell specific TFs have been worked out by numerous prominent B cell labs that the authors are now claiming credit for. I suggest the authors read up on regulatory network inference before making claims of this sort.

Finally, in the Discussion there is no mention of the recent studies performed by the Dekker lab showing changes in chromatin organization during mitosis and no real discussion as to how their finding relate to the finding from Peter Fraser's lab.

Reviewers' comments:

Reviewer #1 (Remarks to the Author):

The manuscript has improved. In my opinion it is nice work and recommend publication in Nature Communication. One minor detail. A recent manuscript describes changes in compartmentalization during plasma cell differentiation (Bortnick et al., 2020). I think this paper should be referenced.

We thank the reviewer for recommending the paper for publication. We had referenced the bioRxiv version of the Bortnick et al paper but have now updated it to cite the Cell Reports version.

Reviewer #2 (Remarks to the Author):

In this revised manuscript the authors have done surprisingly little to address what were substantive criticisms from the reviewers. The most offensive of all was the causality statement in the title of the paper, which suggests causality and an order of events that is completely unsubstantiated by the data. This is unacceptable.

We agree that this is a purely descriptive study, the strength of which lies in the analysis of genome and transcriptional alterations in activation, cell division and differentiation over time in primary cells. We clearly overstretched our language to infer causality around the role of genome organisational changes and transcription and did not appropriately rectify this in the first round of revision. We apologise. We have now made additional changes that remove any such unsubstantiated claims. We hope this addresses the Reviewers valid and serious concerns. Among numerous changes, we have altered the title to “Pre-mitotic genome reorganisation bookends the cellular differentiation process”

Regarding my overall comments:

In response to Point1, the authors did not provide distance decay curves or reproducibility scores. These are not details as the authors claim, but are essential to interpretation of the data.

In response to the Reviewer's original review we added a Supplementary Table containing library details which the reviewer has not found sufficient. While we felt that this information, plus the MDS plots contained within Fig 1, would allow accurate interpretation of the data quality and reproducibility, we now include reproducibility scores and decay curves in Supp Fig 1 G and H as suggested by the reviewer. The reproducibility scores were calculated between all libraries with the framework described in Yang et al, 2017 and implemented in the R package hicrep. This uses the stratum adjusted correlation coefficient (SCC) as a measure for quantifying differences between HiC contact matrices. The results are plotted as a heatmap in Supp Fig G where scores can range from [-1, 1]. Scores close to 1 indicate similarity. Of note, all SCCs between biological replicates are higher than 0.98.

To generate decay curves, the interaction frequency as a function of interaction distance using the discrete binning method (1000 bp) from summed biological replicates were plotted. Plasmablasts show an increase in frequency of interactions >1 Mbp compared to the other cell types followed by a decrease after 30 Mbp, as previously reported (Bortnick et al, 2020).

The authors chose to ignore our Point2, stating that their display of the data “is a standard in the field”, which is simply not true. There are several ways to normalise matrices so that they are comparable between samples, for instance iterative correction normalisation or even show the log2 (observed over expected). Since the authors are comparing between cell stages, this is quite important.

While we firmly believe that our method to allow comparison of contact matrices is sound, we understand having different maximum pixel values for different samples can be confusing for some readers. Thus, as requested, we now have applied the implicit normalisation approach of iterative interaction read balancing (Imakaev et al, Nat Methods 2012) to 50 kbp resolution contact matrices with the HOMER HiC pipeline (Heinz et al, 2010; Heinz et al, Cell, 2018) to all contact matrices shown in the manuscript. Of note, the results of the two methods allow for the same interpretation.

Main points:

Point 1 My criticism, shared by all reviewers and strongly by reviewer #3, was that there is no causal dissection of genome organisational changes and transcriptional changes so such conclusions are unwarranted’. We suggested an experiment that could permit preliminary causal conclusions, which they did not respond to. They claim to have made 'extensive changes' in wording but I could only find small changes that did not revise their interpretation of the data in a way that is consistent with the descriptive nature of the data.

We have further changed our wording to remove the inference of causality throughout the manuscript.

Point 2 The authors performed the analysis we asked for and changed the figure legend accordingly. However, the original manuscript was largely predicated on the fact that changes in architecture only occur prior to mitosis when transcription changes for clonal expansion are established. The results of this new analysis of the same data challenge that conclusion. But the authors do not incorporate this result into their overall conclusions. The authors should at least integrate this new result in their later analysis, showing the fraction of changes that have no gene transcription consequences, etc.

In the first review we reanalysed the TAD size data across all samples to enable P value determination as suggested by this reviewer. This showed that there is was also a significant alteration in TAD size by 3 hours after B cell activation which we acknowledged in the revision by stating “A similar, albeit weaker, pattern of diminishing TADs is also observed in the first 3 hours after B cell activation (Fig 2 B-F), possibly reflecting the transcriptional and cellular responses to activation.” Although there is a change in TAD size at this time point, our DI and DIP analysis found that the genome organisational changes that occurred 3 hours after activation are dwarfed by the large number of alterations between 10-33hrs and between expanded-plasmablasts. Therefore, this does not affect our conclusion that the major changes in genome organisation occur, 1) prior to the first division and are maintained as B cells clonally expand and 2) as expanded cells differentiate into plasmablasts.

Point3 The authors say they made changes but the statements we pointed out remain completely unchanged

We have further altered these statements to remove inference of causality.

....and they did not address any of our criticisms of the figure itself with any revisions.

Changes have been made to Fig 4 to address the Reviewers concern surrounding the number of TFs labelled. Regarding plotting TF ChIP peaks alongside interactivity, we refer the Reviewer our analysis of Irf4 and Blimp1 on page 12.

Point4-5, same issue, adding 'putative' to a phrase does not equate with the careful reassessment of the relationship between form and function.

We have further altered these statements to remove inference of causality.

Minor points:

Point3 We asked for Fig3f to be expanded to include all DIPs not just key genes, and this was ignored.

We did not ignore the reviewer. The data the reviewer was interested in was already present in Supp Fig 2F and we did not think it added to the narrative, interpretation or impact of the work to include it in the main figure. Our original response was:

“3) Fig3e can be accompanied by a similar heatmap but showing all the DIPs. How many of DIPs show this ‘two waves’ of changes? Can Fig3f be expanded to include all DIPs, not just the several selected key b cell genes? We refer the Reviewer to Supplemental Figure 2 F for heatmaps of all DIPs at each transition.”

If deemed necessary by the Editor Supplementary figure 2F can be moved to Fig 3, however, we don't feel its position in the manuscript alters the narrative or impact of the work.

Point4 The authors did not indicate any changes in the manuscript in response to this point. There are indeed a lot of genes that changed expression, more than half of all genes, some of them housekeeping genes... Are they all or mostly accompanied by some sort of interaction changes? If so, the genome organisational changes could very well be a reflection of the changes in transcriptional output instead of the cause of transcription changes specific to differentiation.

Our response to the Reviewers original question (both shown below) was meant only to clarify and highlight where the answers to the Reviewers question could be found within the manuscript.

“4) If you add up all the genes they claim are significantly altered, it amounts to nearly all the annotated genes in the genome. Do this include housekeeping genes? Can the authors explain? Are some going up and then down again perhaps?

If we sum all the DEs detected across all transitions of B cell differentiation we find 12,326 significant transcriptional changes during B cell differentiation. We also detect >17,000

significant changes in genome organization (DIs). Both DEs and DIs include “housekeeping genes” and both include increases and decreases at each transition (see Fig 1 C and E). Changes in expression or structure can be enhanced or reversed at any other transition. Each transition is analysed independently of all others.”

In response to the new extensions to the question posed by Reviewer 2 (eg the expression link to interactivity) we have shown that the bulk of the transcriptional change (Fig 1C) is not associated with organisational change (Fig 1E), as the vast majority of organisational change occurs in two discrete waves, while transcriptional change occurs throughout differentiation. Transcriptional change that occurs during these waves of organisational change are positively associated with the organisational change (Fi 3 G-J).

Reviewer #3 (Remarks to the Author):

Pre-mitotic genome reorganization establishes the transcriptional imprint for cellular differentiation

I was disappointed to find that very little was changed during the revision and the authors continue to make statements that are not supported by their data.

It is clear that we gravely underestimated the gravity of the reviewer’s concerns and failed to appropriately address them. For this we apologise. We have now made many more changes to the manuscript to remove any unsubstantiated claims.

In addition, they fail to adequately discuss their work in the light of papers from the Fraser and Dekker labs who have both done extensive analyses regarding changes in chromatin organization during cell cycle / cell division.

We apologise for the lack of discussion of these previous studies of genome organisation and cell cycle. We have now included substantial mention of these works in both the introduction and discussion and have tried to clarify that our study differs from these because we have studied when in cell cycle and division the genome was restructured in response to activation and differentiation stimuli.

Abstract: These changes reveal an elaborate gene-regulatory network providing an explanation for how lymphocyte fate is imprinted prior to the first division’

Is an overstatement as the authors did not generate a regulatory network. I suggest the authors read up on regulatory network inference before making claims of this sort.

We apologise for overstatement and have now changed this sentence to remove the mention of a gene-regulatory network.

Line 17 – the authors still have not referenced the work of Dekker and Fraser

These references have now been added.

Line 25: Overall, we propose that chromosome reconfiguration is spatiotemporally separated from DNA replication and Mitosis – this not knew and has been shown previously by Dekker and Fraser

We apologise for not making the distinction clearer. We have now altered this sentence to: “Overall, we propose that the 3D genome is reconfigured in response to differentiation signals prior to DNA synthesis and mitosis to ensure the implementation of a transcriptional program required for the generation of B cell immunity.”

Page 6 Line 12: As such, strong three-dimensional connections between the *Twistnb* promoter and putative long-range enhancers are detected prior to the first activation-induced division but not after (Fig 1 F). There is no evidence for these being putative enhancers – if the authors want to make this claim they should overlap with ATAC-seq and/or H3K27ac data at the same time points to show this.

This sentence has been altered to: “As such, increased three-dimensional contacts between the *Twistnb* promoter and distant sites in the genome are detected prior to the first activation-induced division but not after (Fig 1 F).”

Our organizational data suggests many of the changes in genome organization we observe are likely promoter-enhancer interactions.

This sentence has now been removed.

Page 10: ‘hinting that many more transcription factors than previously demonstrated may regulate immune cell gene expression via three-dimensional genome organisation (Hu et al., 2018; Johanson et al., 2018a).

This sentence should be changed to ‘hinting that many more transcription factors than previously demonstrated may regulate immune cell gene expression and these changes are linked to changes in three-dimensional genome organisation’.

This way there is no cause or effect implied.

As requested, we have changed this sentence to: “...hinting that many more transcription factors than previously demonstrated may regulate immune cell gene expression and these changes are linked to alterations three-dimensional genome organisation.”

Page 12: Thus, consistent with our association analyses and those of others, the presence of *Irf4* or *Blimp1* within promoter interacting regions significantly increases the chance that these structures are removed or repressed.

This sentence should be changed to ‘Thus, consistent with our association analyses and those of others, the presence of *Irf4* or *Blimp1* within promoter interacting regions significantly increases the chance that these structures are altered.

We have altered this sentence as suggested by the reviewer.

Page 12: Taken together these analyses suggest that overlays of transcription factor expression data and genome organizational data may be used to infer the intricate and previously undetectable regulatory network of TFs influencing B cell activation and differentiation.

The authors have linked changes in TF expression to changes in chromatin structure – they have definitely NOT inferred a regulatory network of TFs influencing B cell activation. B cell specific TFs have been worked out by numerous prominent B cell labs that the authors are now claiming credit for. I suggest the authors read up on regulatory network inference before making claims of this sort.

We have altered this sentence to: “Taken together these analyses suggest that overlays of transcription factor expression data and genome organizational data may be used to infer the TFs influencing B cell activation and differentiation.”

Finally, in the Discussion there is no mention of the recent studies performed by the Dekker lab showing changes in chromatin organization during mitosis and no real discussion as to how their finding relate to the finding from Peter Fraser’s lab.

We have now added a paragraph in the discussion that specifically refers to previous work on 3D genome and cell cycle. We have also reworded sentences throughout the manuscript to ensure that it is clear that our study is different because it is specifically studying the response to cellular activation and differentiation signals.

REVIEWERS' COMMENTS

Reviewer #3 (Remarks to the Author):

Pre-mitotic genome reorganization establishes the transcriptional imprint for cellular differentiation

The authors have made many changes that I requested but there are still issues that need to be fixed.

In the abstract the last sentence, 'They also suggest that chromosome reconfiguration occurs prior to DNA replication and mitosis and guides a gene expression program that controls the differentiation process

' should be changed to: 'They also suggest that chromosome reconfiguration occurs prior to DNA replication and mitosis and is linked to a gene expression program that controls the differentiation process'. There is no evidence for anything causal here as pointed out in my two previous reviews.

Page 5. The PI discusses changes that occur after B cell activation prior to the first division. Then he leaps to discussing T and B cell fate and the first cell division. What first cell division? This needs to be rewritten and better explained.

Figure 1F and 3A: HiC data should be shown as subtraction data. It is very hard to see any differences at each time point the way the data is displayed.

Page 6 – given that so many genes are altering expression (5838) in the first three hours while there is much less evidence of chromosome reconfiguration – the data suggest in the case of these genes transcription could drive the alterations in chromatin folding. The authors should comment on this and identify precisely how many of the transcriptional changes occur prior to any HiC changes.

Page 7. Referring to the Twistnb gene the authors make the statement that changes in chromatin folding occur within the first wave – how is this linked to expression? Also they state that structure diminishes when the RNA polymerase is no longer required. Do they have evidence to support the fact that RNA PolI is no longer transcribed?

Page 7: The sentence, 'Interestingly, the organizational changes around the Bcl6 gene among others such as Ebf1, Prdm1 and Id2 reflect its expression pattern, suggesting that chromosome structure potentially plays a role in regulating Bcl6 expression.....'

 should be changed to: 'Interestingly, the organizational changes around the Bcl6 gene among others such as Ebf1, Prdm1 and Id2 are linked to their expression'. Again, there is no evidence for a causal effect here.

Page 7: The sentence, 'The first is that given the relative absence of early activation induced genome organizational changes, the rapid and dramatic transcriptional changes that occur immediately post-activation are either driven independently of 3D structure or rely on pre-existing structures', should be changed to: 'The first is that given the relative absence of early activation induced genome organizational changes, the rapid and dramatic transcriptional changes that occur immediately post-activation suggest that transcriptional changes could be driving changes in 3D structure'. This can be checked. How genes whose transcription changes are linked to subsequent changes in 3D structure?

Reviewer #4 (Remarks to the Author):

Chromatin reprogramming during development and differentiation was previously reported. The current submission provides data suggesting that reprogramming of chromatin occurs early in differentiation, prior to the first cell division in B cell differentiation (prior to the first genome duplication that would occur before those cells divide).

The study utilizes appropriate, state of the art methodology that support the conclusions and the data are analyzed with sufficient statistical power. Although the study is primarily descriptive, the role of chromatin modulators during the onset of differentiation is yet to be understood, the work will be of significance to the field. The findings should be discussed in the context of other recent studies dissecting plasma cell differentiation, for example, studies evaluating the effects of LSD1 and EZH2 on chromatin accessibility (e.g. PMID: 30232138, J. Imm 2018; PMID: 29703886, Nat Commun. 2018).

Minor suggestions:

One sentence summary: bookend

Page 4 line 5: change

Johanson 2018b incomplete reference

REVIEWERS' COMMENTS

Reviewer #3 (Remarks to the Author):

Pre-mitotic genome reorganization establishes the transcriptional imprint for cellular differentiation

The authors have made many changes that I requested but there are still issues that need to be fixed.

In the abstract the last sentence, 'They also suggest that chromosome reconfiguration occurs prior to DNA replication and mitosis and guides a gene expression program that controls the differentiation process' should be changed to: 'They also suggest that chromosome reconfiguration occurs prior to DNA replication and mitosis and is linked to a gene expression program that controls the differentiation process' . There is no evidence for anything causal here as pointed out in my two previous reviews.

The requested change has been made in the further revised manuscript.

Page 5. The PI discusses changes that occur after B cell activation prior to the first division. Then he leaps to discussing T and B cell fate and the first cell division. What first cell division? This needs to be rewritten and better explained.

This section has been expanded to better explain the first division *after activation* and the link between the prolonged G1 phase and fate decisions.

Figure 1F and 3A: HiC data should be shown as subtraction data. It is very hard to see any differences at each time point the way the data is displayed.

We don't agree that subtraction data would be helpful or appropriate for these figures. The matrices in Fig 1F show two of >10,000 changes observed during B cell development (Fig 1E) and are included largely to provide an example of the quality and form of our data. Absolute rather than subtracted intensities are necessary to display the structure of the data. In the case of Fig 3A the contact matrices are included simply to provide context. The significant differences are precisely and specifically displayed by the arc plots in the upper panel of Fig 3A.

Page 6 – given that so many genes are altering expression (5838) in the first three hours while there is much less evidence of chromosome reconfiguration – the data suggest in the case of these genes transcription could drive the alterations in chromatin folding. The authors should comment on this and identify precisely how many of the transcriptional changes occur prior to any HiC changes.

~11,000 expression changes occur prior to any major organisational change. Regarding the relationship between these changes and subsequent organisational change, we did examine associations between transcriptional and organisational changes across, as opposed to within, time points. We found no correlations between the two between/across B cell development stages, except for a significant ($P = 2.46 \times 10^{-6}$) association between pre-existing structures and transcriptional change immediately after activation (3hr). This suggests that the act of transcription itself at any time point does not influence genome architecture at another time point. Of further note, our gene ontology analyses provide a logical link between transcriptional change and the organisational change we observe prior to the first post-activation division – showing enrichment of

genes involved in DNA folding etc. only at the 10hr – Imminent division transition. Thus, it appears that transcription can influence organisation, and vice versa (Fig 3 G-J), but mostly within the same stage of development.

Page 7. Referring to the Twistnb gene the authors make the statement that changes in chromatin folding occur within the first wave – how is this linked to expression? Also they state that structure diminishes when the RNA polymerase is no longer required. Do they have evidence to support the fact that RNA Pol is no longer transcribed?

The expression of Twistnb, a component of the RNA polymerase I complex, across B cell stages is shown below. It roughly mirrors the pattern observed in 3D organisation around the gene, with most activity occurring prior to the first division which diminishes as B cell development progresses.

Page 7: The sentence, ‘Interestingly, the organizational changes around the Bcl6 gene among others such as Ebf1, Prdm1 and Id2 reflect its expression pattern, suggesting that chromosome structure potentially plays a role in regulating Bcl6 expression.....’ should be changed to: ‘Interestingly, the organizational changes around the Bcl6 gene among others such as Ebf1, Prdm1 and Id2 are linked to their expression’. Again, there is no evidence for a causal effect here.

The requested change has been made in the further revised manuscript.

Page 7: The sentence, ‘The first is that given the relative absence of early activation induced genome organizational changes, the rapid and dramatic transcriptional changes that occur immediately post-activation are either driven independently of 3D structure or rely on pre-existing structures’, should be changed to: ‘The first is that given the relative absence of early activation induced genome organizational changes, the rapid and dramatic transcriptional changes that occur immediately post-activation suggest that transcriptional changes could be driving changes in 3D structure’. This can be checked. How genes whose transcription changes are linked to subsequent changes in 3D structure?

We have not made the requested change as we disagree with the recommended statement. First, we have shown that pre-existing structures significantly ($P = 2.46 \times 10^{-6}$) correlate with initial (3hr) post-activation transcription. Second, as mentioned above, during our exploratory analyses of the data (and as wisely suggested by the Reviewer), we looked for association of stage-specific transcriptional change with organisation change at other stages of B cell development, both pre- and post-transcriptional change, and find no correlations. This suggests that, apart from the pre-existing structures, transcriptional change and organisational change do not influence each other across

stages of B cell activation and development.

Reviewer #4 (Remarks to the Author):

Chromatin reprogramming during development and differentiation was previously reported. The current submission provides data suggesting that reprogramming of chromatin occurs early in differentiation, prior to the first cell division in B cell differentiation (prior to the first genome duplication that would occur before those cells divide).

The study utilizes appropriate, state of the art methodology that support the conclusions and the data are analyzed with sufficient statistical power. Although the study is primarily descriptive, the role of chromatin modulators during the onset of differentiation is yet to be understood, the work will be of significance to the field. The findings should be discussed in the context of other recent studies dissecting plasma cell differentiation, for example, studies evaluating the effects of LSD1 and EZH2 on chromatin accessibility (e.g. PMID: 30232138, J. Imm 2018; PMID: 29703886, Nat Commun. 2018).

The discussion section of the further revised manuscript has been modified to include a discussion of our work in the context of these previous works.

Minor suggestions:

One sentence summary: bookend

We thank the Reviewer for their attention to detail. The requested change has been made in the further revised manuscript.

Page 4 line 5: change

The requested change has been made in the further revised manuscript.

Johanson 2018b incomplete reference

We thank the reviewer for their outstanding attention to detail and apologise for the oversight. The reference duplication has been corrected.